# Reducing Semantic Mismatch in Brain-to-Text Decoding Through Personalized Multimodal Masking

**Jiaxuan Chen[1,2], Yu Qi[2,3,*] Yueming Wang[1] & Gang Pan[1,2]**
[1]College of Computer Science and Technology, Zhejiang University
[2]State Key Lab of Brain-Machine Intelligence, Zhejiang University
[3]Affiliated Mental Health Center and Hangzhou Seventh People's Hospital, MOE Frontier
Science Center for Brain Science and Brain-Machine Integration, Zhejiang University
`{jiaxuan_chen, qiyu, ymingwang, gpan}@zju.edu.cn`

## Abstract

The rapid progress of large vision-language models (VLMs), such as CLIP, has spurred the development of a wide range of neural decoding frameworks. Nevertheless, most existing approaches still suffer from semantic mismatches during representational alignment. This challenge may stem from the fact that the human brain does not distribute attention uniformly across a visual scene, but rather selectively encodes salient or relevant regions. Moreover, such selectivity is closely related to individual interests and varies from person to person. To address this challenge, we propose Yo'Mind, a novel optimal transport (OT)-driven personalized multimodal semantic masking framework designed to bridge the semantic gap between brain and machines in interpreting visual scenes. Technically, Yo'Mind introduces a dynamic semantic pruning and allocation mechanism that adaptively masks redundant visual semantic components in stimulus images based on individual neural responses—without requiring extra human supervision or hyperparameter tuning. This strategy can be used to enhance semantic consensus between brain and machine representations during decoding. Furthermore, the inherent flexibility of OT theory enables Yo'Mind to perform brain-visual-linguistic alignment and cross-subject decoding within a unified end-to-end architecture. Extensive experiments demonstrate that our Yo'Mind offers several advantages, including state-of-the-art brain-to-text reconstruction performance and improved interpretability of the decoding process.

## 1 Introduction

Artificial intelligence (AI) researchers have long sought to develop systems that understand the physical world in a human-like fashion (Duéñez-Guzmán et al., 2023; Chen et al., 2025b). Recent years have seen incredible strides in artificial neural networks (ANNs) through diverse large-scale datasets and increased model sizes (Radford et al., 2021; Brown et al., 2020). Despite findings from computational neuroscience suggesting that representations of certain large models (LMs) serve as better proxies for interpreting neural responses (Wang et al., 2023), substantial semantic biases still remain between the representations of biological and artificial systems (Chen et al., 2025b; 2024a). ANNs often latch onto statistical patterns present in the static training data (Duéñez-Guzmán et al., 2023), thereby allowing their embeddings to exhaustively encode nearly every visible element in a complex scene. In contrast, humans only selectively attend to those visual-semantic elements deemed important for understanding or interacting with the scene, exhibiting significant individual variability (Chen et al., 2025c; 2023a).

For example, consider a scene in which a young `boy` is flying a `kite` along a scenic `lakeshore`, accompanied by a nearby `dog` (see Fig. 1). While some individuals may focus primarily on the boy and his behaviour, others might be drawn to the tranquil beauty of the surrounding landscape.

---

*Corresponding author.

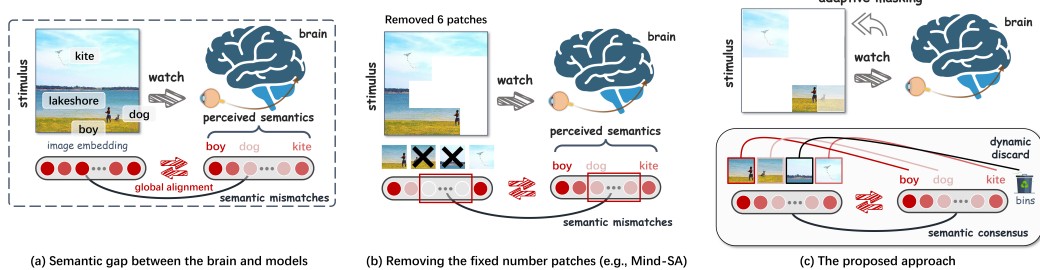

Figure 1: (a) Illustration of semantic biases between brain and model embeddings, and the limitations of global alignment in neural decoding. (b) Semantic consensus preserving by removing a fixed number of image patches. (c) Our proposed Yo'Mind with an OT-driven personalized semantic masking mechanism, designed to support subject-adaptive, fine-grained alignment. Darker (or lighter) colors represent stronger (or weaker) semantic responses.

Therefore, directly aligning image representations with the neural patterns recorded while subjects were viewing the image can be problematic (Chen et al., 2025c) (Fig. 1 a). Such an issue becomes even more pronounced when attempting cross-subject decoding within a single model, as different subjects may interpret the same image in distinct ways. A more promising paradigm is to dynamically detect the visual-semantic elements that the brain actually focuses on and leverage them as supervision to guide semantic reconstruction from brain activity. Typically, during fMRI acquisition, subjects are instructed to maintain central fixation, but fixation at a single point does not necessarily imply uniform semantic attention due to the possibility of covert attentional shifts (Posner, 1980).

Recently, Chen et al. (2025c) have highlighted the semantic mismatch issue in neural decoding, and have introduced a new reconstruction model Mind-SA. The core idea is to preserve semantic consensus between brain activity and image embeddings by removing a fixed number of image patches, as demonstrated in Fig. 1 (b). Although Mind-SA can be trained in an end-to-end manner with the help of differentiable techniques (Jang et al., 2016), it still suffers from several limitations. First, using a fixed number of patches limits generalization, as scenes of varying complexity may require the removal of different amounts of redundant semantic content. Second, hard selection strategies lack fine-grained control. In many cases, a semantic component may be weakly represented in neural signals rather than entirely absent, making it inappropriate to discard it completely.

To address the limitations of existing methods (Takagi & Nishimoto, 2023b; Ferrante et al., 2023; Mai & Zhang, 2023; Xia et al., 2024; Shen et al., 2024; Chen et al., 2025c;a), we propose Yo'Mind, a novel alternative decoding paradigm, to revisit the semantic mismatch issue in neural decoding. Our insight is that we can leverage the optimal transport (OT) framework (Peyré et al., 2019; Cuturi, 2013) to establish an assignment structure between a visual stimulus and its evoked neural responses by minimizing the matching cost, where each stimulus image is viewed as being composed of a set of semantic components. The key here lies in how to deal with semantic components that are not encoded in the neural patterns, without the need for any additional annotations or hyperparameter tuning (*e.g.*, setting a threshold to suppress semantic components).

In Yo'Mind, we augment each set with a virtual dustbin so that unrelated semantic components can be explicitly assigned to it, as shown in Fig. 1 (c). This formulation avoids the use of hard masking and enables a dynamic soft selection of semantic elements, making it highly suitable for fine-grained alignment and cross-subject decoding. Notably, with the help of multimodal large language models (MLLMs), Yo'Mind also extends traditional brain-visual alignment into brain-visual-linguistic alignment by incorporating rich linguistic semantic cues related to the target stimuli. In brief, our contributions can be summarized as follows: **1)** We introduce an OT-driven, personalized semantic masking mechanism, designed to seek semantic consensus between brain signals and machine representations, without requiring any annotations or hyperparameter tuning; **2)** We propose a novel brain semantic decoder, Yo'Mind, which leverages the personalized semantic masking to support subject-adaptive, fine-grained alignment across neural, visual, and linguistic modalities; **3)** We demonstrate that our Yo'Mind outperforms existing methods. Extensive results confirm that it improves the interpretability of neural decoding and achieves state-of-the-art cross-subject decoding performance.

## 2 RELATED WORK

Neural decoding, a solid foundation for brain-computer interfaces, offers a promising pathway for bridging the gap between humans and machines. In recent years, fueled by the success of large vision-language models (VLMs) such as Radford et al. (2021) and generative models (*e.g.*, diffusion models (Rombach et al., 2022)), neural decoding has attracted increasing attention and has been extensively studied. Here, we briefly introduce representative studies most relevant to our Yo'Mind.

**Brain-to-Image Decoding.** To reconstruct (or identify) presented visual stimuli from fMRI, early attempts (Miyawaki et al., 2008; Schoenmakers et al., 2013; Kay et al., 2008; Horikawa et al., 2013) often combined linear regression with handcrafted descriptors. Even though these approaches have achieved satisfactory performance in decoding simple stimuli, they are constrained by oversimplified mappings, resulting in degraded performance when decoding complex natural scenes (Rakhimberdina et al., 2021). Building upon powerful generative models such as stable diffusion (SD) (Rombach et al., 2022), recent works have achieved unparalleled performance in brain-to-image reconstruction. For example, Takagi & Nishimoto (2023a) map fMRI voxels to the latent space of SD using a ridge regression model, and then feed the predicted fMRI embeddings into the frozen decoder of SD to reconstruct images. A concurrent work (Chen et al., 2023b) also used the diffusion model but adopted self-supervised modeling to learn fMRI representations. MindEye (Scotti et al., 2024) incorporates contrastive learning and a diffusion prior for brain-to-image retrieval and reconstruction. Beyond the aforementioned methods, several other representative studies (Chen et al., 2024b; Lu et al., 2023; Chen et al., 2025a) have also explored diffusion models conditioned on brain signals, which have grown to be a dominant paradigm for brain-to-image decoding. The success of this paradigm depends on the alignment between fMRI signals and the target representations, and CLIP has been demonstrated to be a good proxy model (Chen et al., 2024a). More recently, several studies (Scotti et al.; Wang et al., 2024) have also investigated cross-subject decoding, with the central idea being to learn a shared representational space.

**Brain-to-Text Decoding.** Unlike pixel-level reconstruction of stimulus images, brain-to-text decoding focuses on translating brain activity into semantically meaningful word sequences, which is considered a more efficient form of "mind reading" (Chen et al., 2025a). This is because the human brain does not function like a camera that records every pixel (Chen et al., 2023a). Technically, the key to brain-to-text decoding remains the alignment between fMRI signals and the target representational space. The difference is that visual generative models are replaced by large language models (LLMs) such as GPT-2 (Radford et al., 2019). For brain-to-text decoding, early works (Ferrante et al., 2023; Mai & Zhang, 2023; Takagi & Nishimoto, 2023b) also relied on linear regression to bridge the gap between fMRI and model embeddings such as GIT (Wang et al., 2022) and BLIP (Li et al., 2022). Chen et al. (2025a) proposed an end-to-end brain semantic decoder, MindGPT, trained via self-supervised learning. MindGPT achieves performance improvements, but remains limited by the per-subject-per-model paradigm. To enable cross-subject decoding, UMBRAE (Xia et al., 2024) proposed a multi-subject training strategy to map subject-specific features to a shared latent space. On the other hand, Shen et al. (2024) designed a Vision Transformer 3D for learning cross-subject embeddings. MindLLM (Qiu et al., 2025) learned brain representations using a subject-agnostic fMRI encoder with a neuroscience-informed attention mechanism. A recent study, Mind-SA (Chen et al., 2025c), was the first to highlight the semantic mismatch problem during decoding, and addressed it by removing a fixed number of image patches. Our Yo'Mind goes one step further, and introduces a dynamic semantic pruning and allocation mechanism, thus enabling fine-grained multimodal alignment across multiple subjects.

## 3 METHOD

Unlike existing global alignment approaches (Chen et al., 2025a; 2024b; Shen et al., 2024; Ferrante et al., 2023; Mai & Zhang, 2023; Takagi & Nishimoto, 2023b;a), Yo'Mind is a unified cross-subject decoding architecture that performs fine-grained alignment across neural, visual and linguistic modalities (see Figs. 2 and 3). During multimodal alignment, Yo'Mind adaptively filters out redundant semantics that are not perceived by the subjects, thereby avoiding the semantic mismatch problem. We introduce the details below.

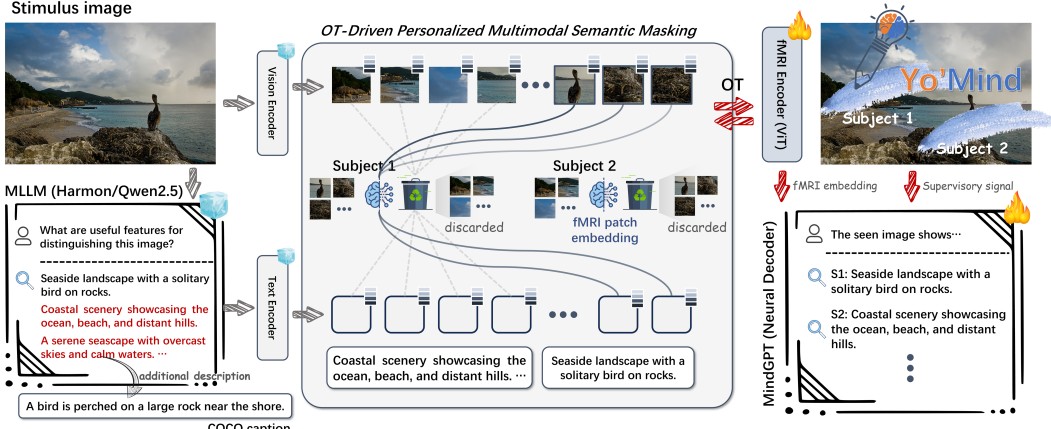

Figure 2: Overview of the proposed personalized multimodal semantic masking mechanism. We first construct a multimodal semantic set for each stimulus image, where the visual semantics are derived from image patches, while the textual semantics are generated by Harmon (Qwen2.5) (Wu et al., 2025). Both the visual and textual data are fed into CLIP to extract representations. Meanwhile, we split fMRI signals into fixed-size patches, and feed the resulting sequence of vectors to a standard ViT. Then, we compute the least costly transport plan under the OT framework, during which irrelevant semantic elements are automatically masked.

### 3.1 CONSTRUCTION OF FINE-GRAINED SEMANTIC SETS

People live in an ever-changing environment full of objects, which shapes our perception to be both directed and selective (*i.e.*, attention). As a result, different people may interpret the same visual scene in different ways (Chen et al., 2025c). To incorporate this mechanism into neural decoding models, the first necessary step is to construct a fine-grained semantic set for each stimulus image.

Formally, given an image $x$, we first reshape $x$ into a sequence of non-overlapping patches, and use a frozen CLIP vision encoder to generate $N$ patch embeddings $\{v_i\}_{i=1}^N$. Meanwhile, we use Harmon (Wu et al., 2025), an advanced multimodal large language model (MLLM), to produce a detailed semantic description of $x$ using the fixed prompt: "*What are useful features for distinguishing this image?*". Generally, MLLMs can produce a rich set of discriminative semantic attributes that capture fine-grained object details, contextual dependencies, and even nuanced scene atmosphere, as shown in Fig. 2 Left. Similarly, we use the CLIP text encoder to produce a set of word embeddings $\{t_i\}_{i=1}^M$ from the semantic description.

### 3.2 OT-DRIVEN PERSONALIZED SEMANTIC MASKING

Once semantic sets for stimuli are constructed, our goal is to determine which semantic elements are truly perceived by the brain. To this end, we propose an OT-driven personalized semantic masking module (see Fig. 2), which produces a partial assignment matrix $\mathbf{P} \in [0,1]^{K \times (N+M)}$. Here, $K$ denotes the number of fMRI patch embeddings. Following Chen et al. (2025a;c), we use a standard ViT (Dosovitskiy et al., 2021) as fMRI encoder.

The OT problem involves finding a transport plan $\mathbf{\Gamma}$ that transforms one probability distribution into another at minimal cost (Peyré et al., 2019). We use the pairwise cosine distances between the fMRI embeddings and the semantic embeddings to construct the cost matrix $\mathbf{C}$:

$$\mathbf{C}_{i,j} = 1 - <r_i, s_j>, \ s_j \in \{v_1, \cdots, v_N, \cdots, t_M\}, \tag{1}$$

where $r_i$ is the fMRI patch embedding, and $< \cdot, \cdot >$ denotes the inner product. Next, we can use the Sinkhorn algorithm (Cuturi, 2013) to solve the entropy-regularized optimal transport problem $\min_{\mathbf{\Gamma} \in \Pi} \langle \mathbf{\Gamma}, \mathbf{C} \rangle - \varepsilon H(\mathbf{\Gamma})$, which is a strictly convex optimization. Here, $\min_{\mathbf{\Gamma} \in \Pi}$ represents finding a transport plan $\mathbf{\Gamma}$ that minimizes the cost, and $\Pi$ denotes the set of all joint distributions whose marginals match the given source and target distributions. $H(\mathbf{\Gamma}) = -\sum_{ij} \mathbf{\Gamma}_{ij} log \mathbf{\Gamma}_{ij}$ is an entropy

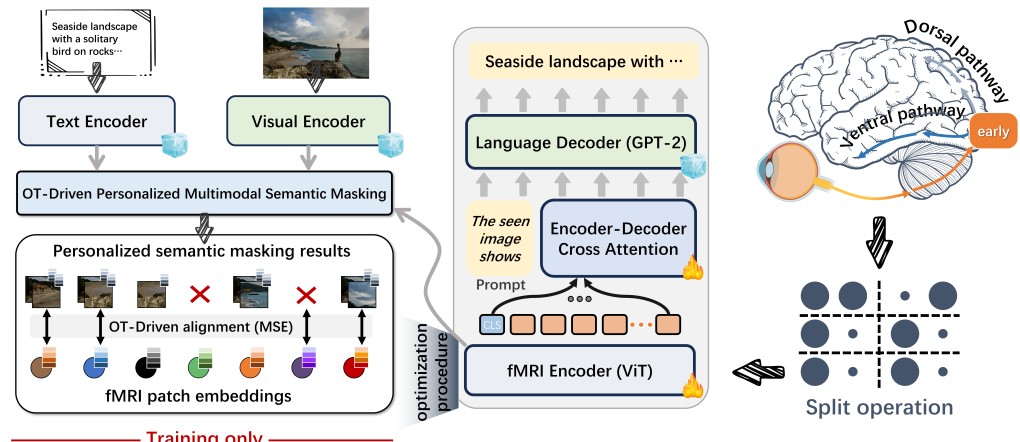

Figure 3: Our end-to-end Yo'Mind architecture. During training, the resulting pruned semantic set serves as a supervision signal for brain-to-text reconstruction.

constraint term and $\varepsilon > 0$ is a trade-off hyperparameter. However, in OT framework, transport plan $\mathbf{\Gamma}$ is restricted to

$$\mathbf{\Gamma}\mathbf{1}_{N+M} = \mathbf{1}_K \ \text{ and } \ \mathbf{\Gamma}^{\mathrm{T}}\mathbf{1}_K = \mathbf{1}_{N+M}, \tag{2}$$

where $\mathbf{1}_K$ represent an $K$-dimensional vector with all entries equal to 1. Inspired by graph matching technique (Sarlin et al., 2020; DeTone et al., 2018), we add each set with a virtual dustbin to explicitly mask redundant elements in the semantic set. Specifically, we augment the cost matrix $\mathbf{C}$ to $\mathbf{C}^*$ by appending a extra row and column, filled with a learnable parameter. Under the circumstances, semantic elements will be assigned to fMRI patches or the dustbin. After Sinkhorn iterations, we can recover a partial assignment matrix from the transport plan $\mathbf{\Gamma}$, i.e., $\mathbf{P} = \mathbf{\Gamma}_{1:K,1:(N+M)}$ by dropping the dustbins. The $\mathbf{P}$ now has the constraints:

$$\mathbf{P}\mathbf{1}_{N+M} \leq \mathbf{1}_K \ \text{ and } \ \mathbf{P}^{\mathrm{T}}\mathbf{1}_K \leq \mathbf{1}_{N+M}. \tag{3}$$

### 3.3    THE YO'MIND ARCHITECTURE

**Brain-Visual-Linguistic Alignment.** Different from traditional representation-level alignment, we use only the semantic elements that may be of interest to human brains as supervision signals (see Fig. 3). Given that CLIP has aligned visual and textual modalities into a shared embedding space, our optimization goal is to minimize weighted mean squared error loss, which can be formulated as

$$\mathcal{L}_{align} = \sum_{i=1}^{K} ||r_i - \sum_{j=1}^{N+M} \mathbf{P}_{i,j} s_j||_2^2, \tag{4}$$

where $\mathbf{P}$ denotes the partial assignment matrix between $r = \{r_1, \cdots, r_K\}$ and $s = \{v_1, \cdots, v_N, t_1, \cdots, t_M\}$. Note that redundant semantic elements in $s$ have been masked during the alignment process, as the dustbins are dropped.

**Brain-to-Text Reconstruction.** With the help of Sinkhorn algorithm, both operations in Yo'Mind are differentiable. Therefore, our method can be seamlessly integrated into off-the-shelf brain-to-text reconstruction frameworks in an end-to-end fashion. To ensure a fair comparison, we follow Chen et al. (2025c) and adopt MindGPT as the brain-to-text module of our method. MindGPT is an autoregressive language model conditioned on brain signals, built upon the GPT-2 architecture (Radford et al., 2019). MindGPT connects an fMRI encoder and GPT-2 through a cross-attention mechanism (Vaswani et al., 2017). Therefore, integrating our Yo'Mind with MindGPT simply involves replacing its original encoder with our OT-guided fMRI encoder $\mathbf{F}(\cdot)$, as shown in Fig. 3. By doing so, the brain-to-text reconstruction can be optimized with the following loss

$$P(\mathcal{W}) = -\sum_{i=1}^{n} \log P(w_i | [w_j]_{j<i}, \mathbf{F}(y); \theta), \tag{5}$$

| Model | Dataset | Subject | Language Similarity Metrics ↑ | | | | | | | |
|---|---|---|---|---|---|---|---|---|---|---|
| | | | B@1 | B@2 | B@3 | B@4 | METEOR | ROUGE | CIDEr | SPICE |
| UniBrain (Mai & Zhang, 2023) | NSD | 1 | - | - | - | - | 16.90 | 22.20 | - | - |
| Brain Captioning (Ferrante et al., 2023) | NSD | 1, 2, 5, 7 | 54.82 | 35.14 | 21.96 | 14.04 | 16.24 | 40.39 | 38.86 | 8.84 |
| Takagi *et al.* (Takagi & Nishimoto, 2023b) | NSD | 1, 2, 5, 7 | 35.22 | 16.37 | 7.22 | 3.30 | 9.77 | 24.61 | 13.52 | 4.88 |
| MindGPT (Chen et al., 2025a) | NSD | 1, 2, 5, 7 | 56.91 | 36.95 | 23.41 | 15.87 | 20.04 | 38.41 | 43.56 | 10.97 |
| Neuro2Language (Shen et al., 2024) | NSD | 1, 2, 5, 7 | 57.19 | 37.17 | 23.78 | 15.85 | 18.60 | 36.67 | 49.51 | 12.39 |
| UMBRAE* (Xia et al., 2024) | NSD | 1, 2, 5, 7 | 57.38 | 37.98 | 25.18 | 17.11 | 18.27 | 41.99 | 51.68 | 11.82 |
| UMBRAE (Xia et al., 2024) | NSD | 1, 2, 5, 7 | 59.09 | 40.13 | 27.05 | 18.40 | 19.24 | **43.64** | **57.76** | **12.42** |
| MindArt (Chen et al., 2024b) | NSD | 1, 2, 5, 7 | 57.26 | 38.62 | 25.17 | 17.71 | 21.93 | 42.86 | 45.95 | 12.27 |
| Mind-SA (Chen et al., 2025c) | NSD | 1, 2, 5, 7 | 59.76 | 39.81 | 27.05 | 18.86 | 38.08 | 42.72 | 54.03 | 12.25 |
| Yo'Mind (Ours) | NSD | 1, 2, 5, 7 | **61.81** | **42.75** | **29.76** | **20.88** | **38.40** | 43.63 | 56.07 | 12.42 |
| Neuro2Language[‡] (Shen et al., 2024) | NSD | 1, 2, 5, 7 | 64.26 | 51.44 | 47.70 | 32.17 | 20.41 | 52.61 | **83.94** | 18.27 |
| MindArt[†] (Chen et al., 2024b) | NSD | 1, 2, 5, 7 | 64.56 | 51.57 | 46.35 | 31.90 | 37.54 | 50.01 | 78.35 | 18.21 |
| Mind-SA[†] (Chen et al., 2025c) | NSD | 1, 2, 5, 7 | 64.78 | **53.54** | 46.38 | 32.46 | 36.25 | 54.26 | 79.48 | 18.72 |
| Yo'Mind[†] (Ours) | NSD | 1, 2, 5, 7 | **65.19** | 53.53 | **48.46** | **33.36** | **39.25** | **55.19** | 81.16 | **19.25** |

Table 1: Quantitative results of brain-to-text reconstruction on the NSD. Here, UMBRAE* is trained on data from a single subject, whereas UMBRAE is trained using data from all four subjects. [‡] and [†] indicate results obtained with BLIP2- and Harmon (Qwen2.5)-generated captions as ground truth, respectively. Some decoding results are derived from Xia et al. (2024); Chen et al. (2025c). The best and second-best results are highlighted using **Bold** and Underline, respectively.

where $y$ denotes the fMRI signals, $\theta$ are the learnable parameters of cross-attention layer in MindGPT, and $\mathcal{W} = [w_i]_{i=1}^n$ indicates the reconstructed word sequence conditioned on the fMRI patch embeddings $r = \{r_1, \cdots, r_K\}$.

## 4 EXPERIMENTS

### 4.1 IMPLEMENTATION DETAILS

The Yo'Mind framework comprises three encoders: a vision encoder, a text encoder, and an fMRI encoder. Specifically, the vision and text encoders are initialized from a pre-trained CLIP model (ViT-B/32), and their parameters remain frozen throughout training. The fMRI encoder is implemented as a standard ViT with 16 layers and 16-head self-attention. The input fMRI signals are flattened into a 1D sequence and uniformly divided into $K = 8$ non-overlapping patches, which are then fed into the transformer encoder. Note that fMRI voxels are selected from the ventral, lateral, and parietal streams, as well as the early visual cortex (27,638 voxels). We adopt the frozen GPT-2$_{\text{Base}}$ for brain-to-text reconstruction. To connect the fMRI encoder and the GPT-2 decoder, each of the 12 layers of the GPT-2 decoder incorporates a 12-head cross-attention layer, where the dimensionality of the projection matrices is set to 4. Pre-trained CLIP and GPT-2 are publicly available on Hugging-Face (Wolf et al., 2020). During the training phase, we perform 100 Sinkhorn iterations to obtain assignment matrix ($\varepsilon = 1$), and only the parameters of the fMRI encoder and the cross-attention layers are learnable. We use the Adam optimizer (Kingma & Ba, 2014) ($\beta_1 = 0.9, \beta_2 = 0.999$) with a learning rate of 1e-4 and a weight decay of 1e-4, maintained until the model converges. Yo'Mind is trained on data from four subjects (see details below) and implemented in PyTorch using four NVIDIA GeForce RTX 3090 GPUs.

### 4.2 DATASETS AND EVALUATION METRICS

**Datasets.** In this paper, the Natural Scenes Dataset (NSD) (Allen et al., 2022) is used to benchmark the proposed Yo'Mind framework. NSD is a high-quality dataset that provides high-resolution fMRI recordings collected from eight subjects over 30–40 sessions using a 7-Tesla scanner. Each subject viewed 10,000 natural images, each repeated three times, selected from the Microsoft COCO dataset (Lin et al., 2014) and cropped to $425 \times 425$ (if needed). Following the prevailing practice (Takagi & Nishimoto, 2023a; Shen et al., 2024; Chen et al., 2025c), our experiments use 27,750 trials from subjects 1, 2, 5, and 7. Among them, 2,770 trials, including 982 COCO images, are used as the test set, while the remaining trials are used as the training set. Note that the 982 test stimuli are shared among all four subjects.

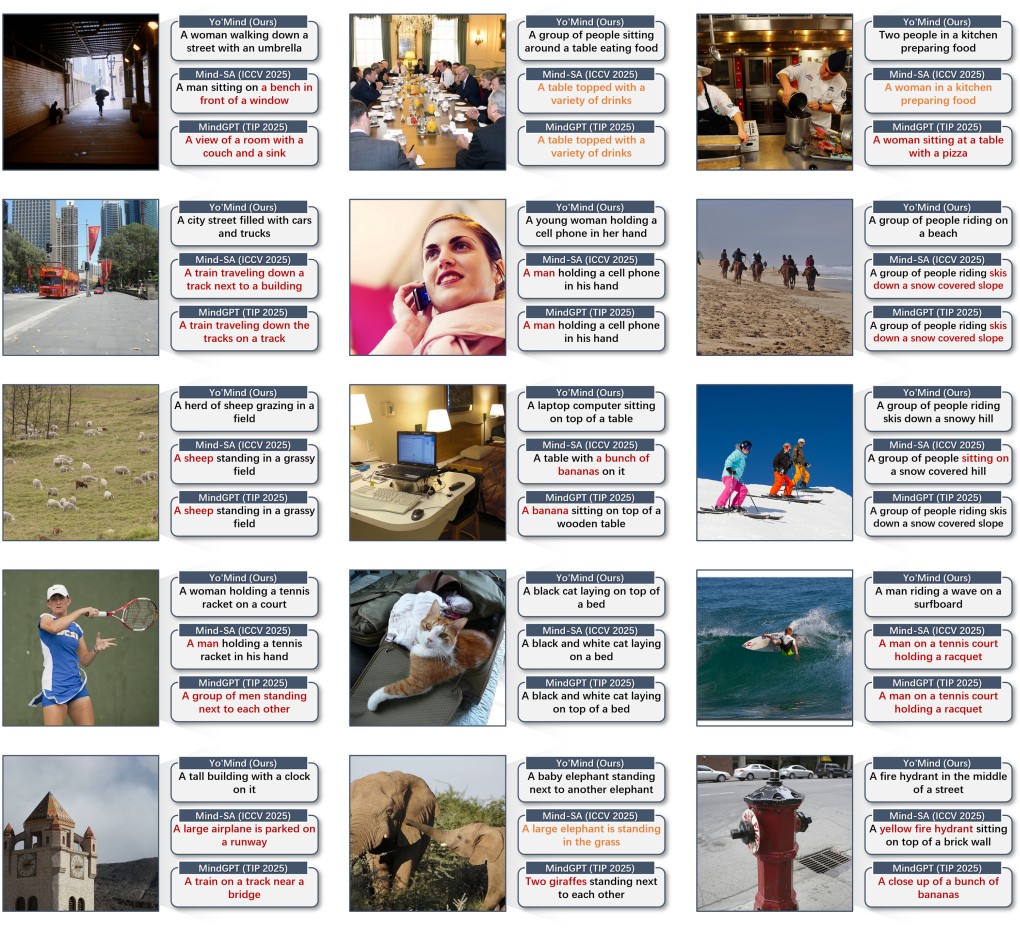

Figure 4: Qualitative results of brain-to-text reconstruction for one subject (S1). For each group, the image on the left is the ground truth stimulus. Imperfect captions are highlighted in *Orange*, and semantic errors are marked in *Red*.

**Evaluation Metrics.** Following previous works (Shen et al., 2024; Chen et al., 2025a;c), we evaluate brain-to-text performance using the following standard metrics: BLEU-1 (B@1), BLEU-2 (B@2), BLEU-3 (B@3), BLEU-4 (B@4) (Papineni et al., 2002), METEOR (Denkowski & Lavie, 2014), ROUGE (Lin & Hovy, 2003), CIDEr (Vedantam et al., 2015) and SPICE (Anderson et al., 2016). These metrics are commonly used to evaluate both n-gram overlap and semantic relevance between generated and reference texts.

## 4.3 COMPARISON WITH STATE-OF-THE-ART

**Qualitative Analysis.** First of all, we qualitatively analyze brain-to-text reconstruction to illustrate the behavioral patterns and performance gains of our Yo'Mind, in comparison with the most relevant existing approaches, Mind-SA (Chen et al., 2025c) and MindGPT (Chen et al., 2025a). Qualitative results are shown in Fig. 4. Intuitively, we observe that Yo'Mind is able to effectively capture and summarize the details of the entire visual scene. In contrast, Mind-SA and MindGPT produce imperfect text reconstructions, such as semantic mismatches or incomplete descriptions.

For example, we see that Yo'Mind can accurately infer the detail "`a woman walking down a street with an umbrella`", even though the visual content appears at a small scale in the image (col 1, row 1). Although Mind-SA fails to reconstruct the main semantic content, it nevertheless captures peripheral details "`a man sitting`". In contrast, MindGPT yields a completely inaccurate reconstruction. In the second case (col 2, row 1), although the image exhibits high visual complexity, Yo'Mind is capable of generating a satisfactory semantic reconstruction

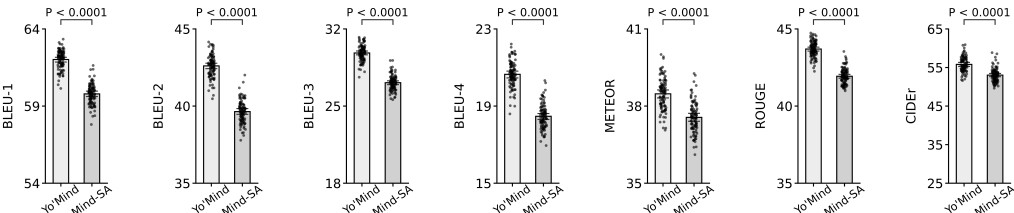

Figure 5: Statistical analysis of model performance. Yo'Mind significantly outperforms Mind-SA across all metrics (two-tailed paired *t*-test, $P < 0.0001$; error bars denote 99% confidence intervals).

that encompasses most key details such as "`a group of people`", "`sitting around a table`", and "`eating food`". By contrast, alternative methods manage to reconstruct only partial semantic content, *i.e.*, "`table`" and "`food`". Other similar cases include Yo'Mind capturing "`a city street filled with cars and trucks`" (col 1, row 2), and "`a baby elephant standing next to another elephant`" (col 2, row 5). These preliminary findings highlight the robustness of Yo'Mind in generating coherent and semantically faithful descriptions, significantly reducing issues of semantic mismatches.

**Quantitative Comparison.** In this part, to enable a comprehensive evaluation, we quantitatively compare the brain-to-text reconstruction performance of Yo'Mind against a wide range of baselines, including Mai & Zhang (2023); Ferrante et al. (2023); Takagi & Nishimoto (2023b); Chen et al. (2025a); Shen et al. (2024); Xia et al. (2024); Chen et al. (2025c). The reconstruction results on NSD (S1, S2, S5, and S7) are summarized in Tab. 1. As shown in Tab. 1, we find that Yo'Mind achieves the highest performance across six metrics when compared with 8 state-of-the-art methods.

Compared to the MindGPT baseline, which can be considered a vanilla method without a semantic pruning strategy, Yo'Mind achieves substantial performance gains: 91.6% on METEOR, 31.6% on BLEU-4, and 28.7% on CIDEr. Importantly, our method consistently outperforms recent Mind-SA, demonstrating a clear and substantial advantage. In general, these quantitative results show that captions produced by Yo'Mind achieve higher quality descriptions, which further demonstrates the effectiveness of the personalized multimodal semantic masking module. Beyond the COCO caption (Lin et al., 2014), we also reran the reconstruction experiments using Harmon (Qwen2.5)-generated captions (Wu et al., 2025), which provide more detailed description of images, as ground truth (see 1 Bottom). The results demonstrate superior performance of Yo'Mind in decoding fine-grained text, suggesting that it can produce not only simple captions but also rich, detailed descriptions. We also provide additional qualitative comparisons on DIR dataset (Shen et al., 2019) in the Appendix.

**Statistical Analysis.** To evaluate whether the performance gains of Yo'Mind are statistically significant, we conduct a statistical analysis on BLEU, METEOR, ROUGE, and CIDEr scores using bootstrap resampling and a two-tailed paired *t*-test. Specifically, we perform 100 bootstrap resamples of the test set and recomputing the metrics for both Yo'Mind and Mind-SA on each resample (see Fig. 5). Results show that Yo'Mind consistently outperforms Mind-SA, and the improvements are statistically significant ($P < 0.0001$ across all evaluation metrics).

## 4.4 REVEALING BRAIN-PREFERRED SEMANTICS

The success of Yo'Mind in brain-to-text decoding can be attributed to the appropriate modeling of brain-preferred semantic elements. To better understand how the proposed OT-driven semantic masking module enhances brain decoding, we investigate its role from two perspectives: visualization of salient regions and comparison with eye-tracking data.

**Visualization of Salient Regions.** As illustrated in Fig. 6 a, we visualize salient regions by projecting the partial assignment matrix of the fMRI [CLS] embedding onto the image patches. We observe that in relatively simple images, which typically contain single-object scenes, the brain-preferred semantic elements are highly consistent. In the first two examples, the preferred semantic content is consistently concentrated on the "`cat`" and the "`skier`" (from left to right are subjects 1, 2, 5, and 7). In contrast, for complex visual images, the brain-preferred semantic content exhibits clear

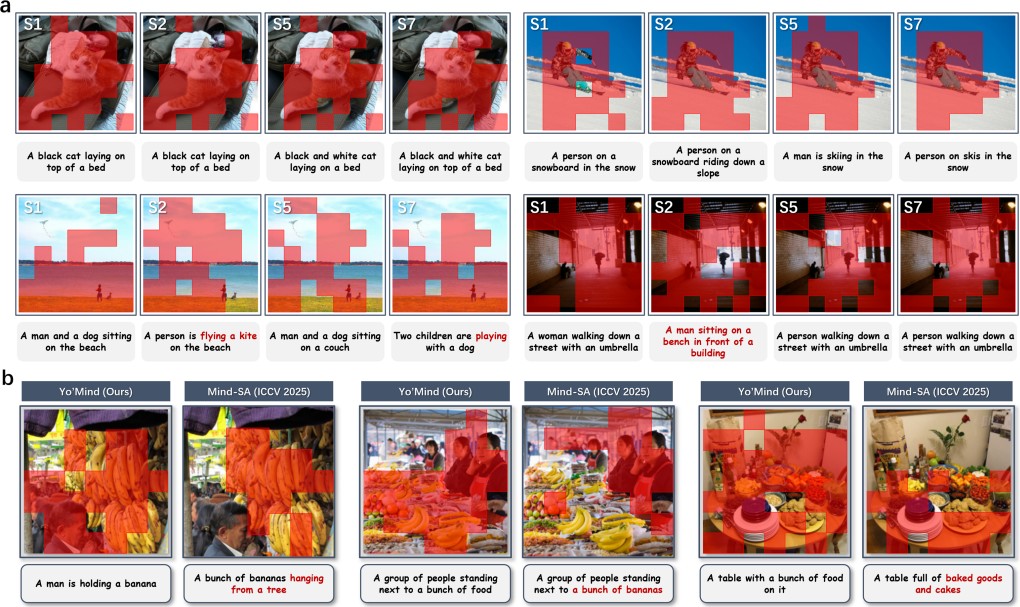

Figure 6: a. The role of the OT-driven semantic masking mechanism. The first row shows examples of intra-subject consistency, and the second row presents examples of inter-subject variability. Only the top 80% salient attention patches are highlighted for visualization, and reconstruction differences are marked in *Red*. b. Comparison of different masking mechanisms. Notably, the hard masking mechanism limits the generalization ability of Mind-SA (Chen et al., 2025c).

| Subject | ROIs | Voxel number | Masking | Language Similarity Metrics ↑ | | | |
|---------|------|--------------|---------|-------|--------|-------|-------|
| | | | | B@4 | METEOR | ROUGE | CIDEr |
| 1 | ventral | 8,590 | ✓ | 17.64 | 35.99 | 41.88 | 46.24 |
| 1 | ventral | 8,590 | ✗ | 15.90 | 21.05 | 38.98 | 43.68 |
| 1 | ventral, lateral | 17,223 | ✓ | 19.42 | 37.68 | 43.41 | 52.80 |
| 1 | ventral, lateral | 17,223 | ✗ | 16.58 | 22.36 | 39.90 | 45.47 |
| 1 | ventral, lateral, parietal | 21,721 | ✓ | **21.78** | **38.98** | **45.82** | **58.87** |
| 1 | ventral, lateral, parietal | 21,721 | ✗ | 16.37 | 21.26 | 38.06 | 46.87 |

Table 2: Effects of brain regions and the semantic masking module on reconstruction performance.

variability. For example, in the third case, subject 2 appears to focus on the action of "`flying a kite`", whereas subjects 1, 5, and 7 concentrate on the "`boy`" and the "`dog`". Finally, we present several examples to illustrate the superiority of our method over Mind-SA. As shown in Fig. 6 b, Mind-SA fails to effectively process complex images, mainly because its fixed image patch masking restricts the model's ability to capture rich visual-semantic cues.

**Comparison with Eye-Tracking Data.** Using eye-tracking to verify whether Yo'Mind truly captures subjects' visual attention may appear to be a straightforward strategy. However, in NSD, all subjects were instructed to maintain central fixation most of the time Allen et al. (2022). As a result, the recorded gaze locations are tightly clustered near the central fixation point and do not reflect the attention when viewing the images. Therefore, we only select the limited cases where the recorded eye positions deviated noticeably from the central fixation and examined their OT-driven semantic allocation plans (see Appendix Fig. A7). Qualitative analysis provides preliminary evidence that Yo'Mind is indeed sensitive to subjects' actual viewing behavior.

It is important to note that gazing at a fixed point does not necessarily imply uniform semantic attention. A large body of work in psychology and neuroscience, such as Posner's covert attention paradigm (Posner, 1980), has shown that subjects can flexibly shift high-level semantic attention without moving their eyes. Validating this would require new datasets that explicitly measure covert attention, which will be an important direction for our future work.

| Subject | ROIs | Voxel number | Visual | Text | Language Similarity Metrics ↑ | | | |
|---|---|---|---|---|---|---|---|---|
| | | | | | B@4 | METEOR | ROUGE | CIDEr |
| 1 | ventral, lateral, parietal | 21,721 | ✓ | ✗ | 21.45 | 38.07 | 44.97 | 57.24 |
| 1 | ventral, lateral, parietal | 21,721 | ✗ | ✓ | 18.35 | 32.95 | 41.87 | 45.02 |
| 1 | ventral, lateral, parietal | 21,721 | ✓ | ✓ | **21.78** | **38.98** | **45.82** | **58.87** |

Table 3: Effects of multimodal semantic set on reconstruction performance.

| Subject | ROIs | Voxel number | Visual | Text | Language Similarity Metrics ↑ | | | |
|---|---|---|---|---|---|---|---|---|
| | | | | | B@4 | METEOR | ROUGE | CIDEr |
| 1 | ventral, lateral, parietal | 21,721 | CLIP | SMALLCAP | 21.36 | 38.14 | 44.56 | 57.21 |
| 1 | ventral, lateral, parietal | 21,721 | CLIP | BLIP3o | **22.23** | 38.46 | **45.97** | 58.47 |
| 1 | ventral, lateral, parietal | 21,721 | CLIP | Harmon | 21.78 | **38.98** | 45.82 | **58.87** |

Table 4: Effects of image captions generated from different models on reconstruction performance.

## 4.5 ABLATION STUDIES

Here we explore effects of different components or configurations on the performance of Yo'Mind.

**OT-Driven Masking Strategy.** First, we explore the effects of personalized multimodal semantic masking on the decoding, as well as investigate the performance across different brain regions (ROIs). Tab. 2 presents the results of Yo'Mind, conditioned on three ROI combinations, with and without the semantic masking strategy. Results show that OT-driven semantic masking significantly enhances decoding performance when using voxels from the ventral, lateral, and parietal streams.

A possible explanation is that different brain regions encode different semantic information. For example, the ventral stream ("what" pathway) exhibits relative specialization in object recognition such as shape and category, whereas the dorsal stream ("where" pathway) is mainly involved in processing spatial location and motion-related information. In other words, different brain regions (voxels) exhibit distinct semantic selectivity. Under this condition, the OT-based semantic masking strategy can assign different semantic elements to different brain areas, thus enabling region-specific alignment (precise matching of specific voxel subsets to corresponding semantic elements).

**Multimodal Alignment.** Next, we report the impact of multimodal (image and text) alignment on the reconstruction performance, as summarized in Tab. 3. The results show that the text description facilitates brain–machine alignment, as evidenced by improved reconstruction performance, which is in line with previous findings such as Du et al. (2023).

**Image Captioning.** Beyond the above analysis, we further evaluate how the quality of image descriptions affects the performance of Yo'Mind. In addition to Harmon, we leverage another advanced multimodal large language model, BLIP3o (Chen et al., 2025d), as well as SMALLCAP (Ramos et al., 2023), a representative classical image captioning model, to provide textual supervision during the alignment process (see Tab. 4). We find that using captions generated by Harmon and BLIP3o as supervisory signals yields better reconstruction performance, suggesting that richer and more structured textual semantics lead to more effective brain decoding.

## 5 CONCLUSION

Our Yo'Mind introduces a novel OT-driven, self-adaptive supervision scheme that reshapes the decoding target space to preserve semantic consensus between brain and machine representations during brain-to-text reconstruction, without the need for any extra annotations or hyperparameter tuning. From the perspective of mitigating semantic bias, Yo'Mind enhances the recent Mind-SA model via personalized multimodal semantic masking. Technically, we use OT tools to avoid fixed hard masking strategies, enabling our Yo'Mind to automatically adapt to visual scenes of varying complexity. In addition, the flexibility of OT allows our method to be seamlessly extended to multimodal alignment and cross-subject decoding. On the other hand, our method holds promise as a neuroscience tool for analyzing intra-subject consistency and inter-subject variability. Nonetheless, such analysis still requires reliable biological validation. This is also a potential limitation of our work. In future studies, we plan to incorporate additional physiological signals, such as eye-tracking, to further validate the brain-preferred semantic elements decoded by our Yo'Mind.

## ETHICS STATEMENT

This study relies entirely on publicly available fMRI datasets (NSD and DIR) and does not involve the collection of any new neural recordings. Our analyses use no personally identifiable information. We follow dataset usage policies and established ethical guidelines to ensure responsible application of brain decoding methods.

## ACKNOWLEDGMENTS

This work was supported in part by the Science and Technology Innovation 2030 Major Projects (No. 2021ZD0200400), Zhejiang Provincial Natural Science Foundation of China (No. LR24F020002), Natural Science Foundation of China (NSFC) (No. 624B2127), and the Qiushi Rising Star Program of Zhejiang University.

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

# A APPENDIX

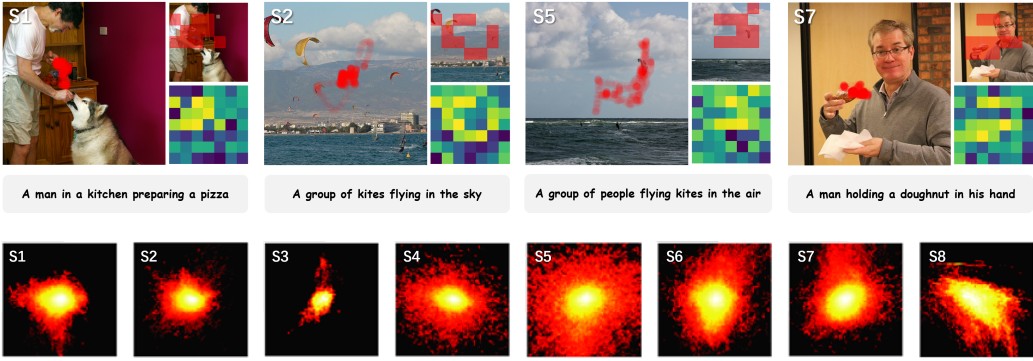

Figure A7: Top: Comparison with eye-tracking. For each group, we visualize the transport plan weights and highlight the top 10 patches with the highest weights. Note that the captions here are reconstructed from Yo'Mind model. Bottom: 2D histograms of gaze positions (log scale) for all subjects in NSD. These results are derived from Allen et al. (2022).

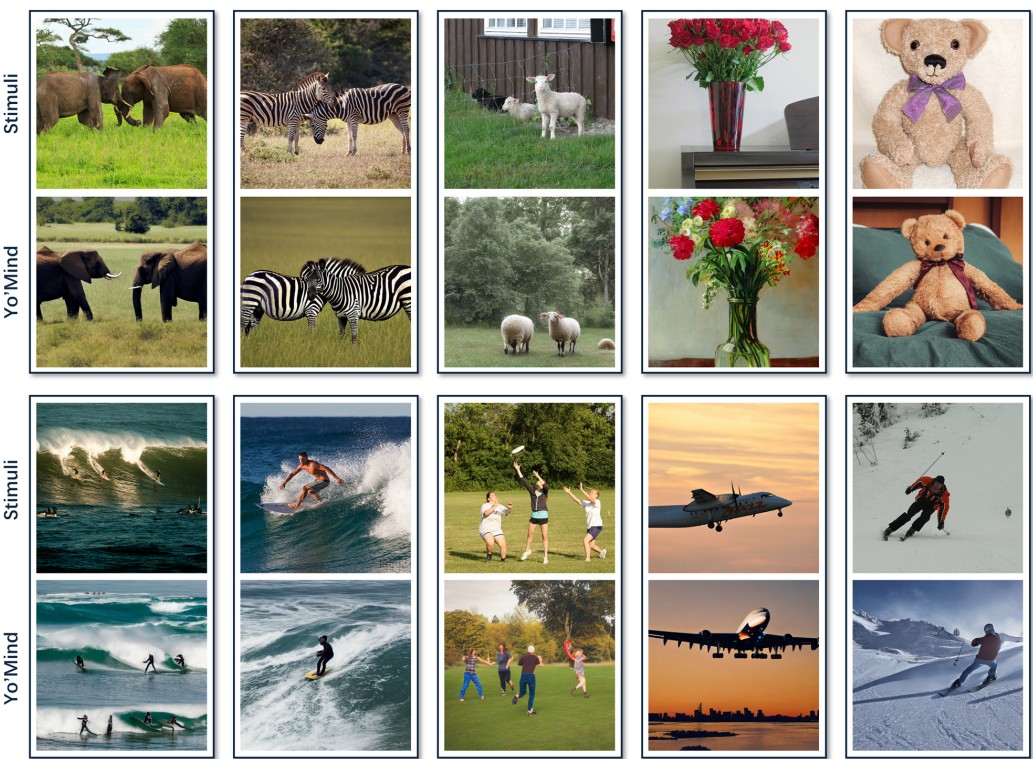

Figure A8: Image reconstruction examples of Yo'Mind.

## A.1 EFFECTS OF THE NUMBER OF FMRI PATCHES

In this study, we divide the fMRI voxels into 8 patches, following the underlying cortical organization of the NSD dataset: early visual, midventral, ventral, midlateral, lateral, midparietal, and parietal regions, with the early visual cortex naturally spanning two patches due to its large number

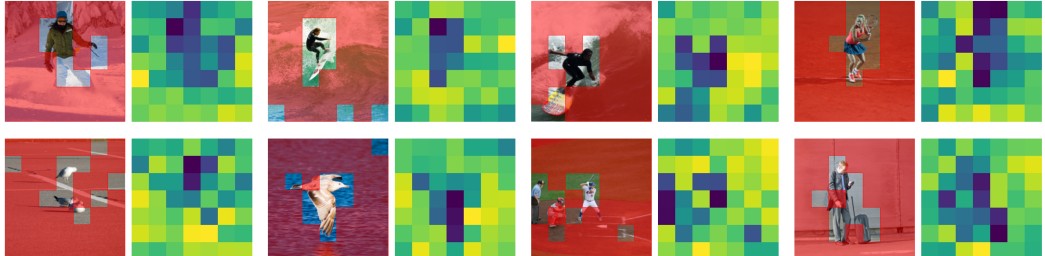

Figure A9: Visualization of the OT plan for the early visual cortex. On the left of each example, we highlight the top 80% of the mass.

| Dataset | ROIs | Number | Language Similarity Metrics ↑ | | | |
|---|---|---|---|---|---|---|
| | | | B@4 | METEOR | ROUGE | CIDEr |
| NSD | early, midventral, ventral, midlateral, lateral, midparietal, parietal | 4 | 17.96 | 34.99 | 40.13 | 52.78 |
| NSD | early, midventral, ventral, midlateral, lateral, midparietal, parietal | 8 | 20.88 | 38.40 | 43.63 | 56.07 |
| NSD | early, midventral, ventral, midlateral, lateral, midparietal, parietal | 16 | 20.74 | 37.73 | 42.47 | 55.14 |

Table A5: Effects of the number of fMRI patches on reconstruction performance.

of voxels. To ensure that each patch has the same dimensionality, we pad the voxel vectors with zeros where necessary. The OT-based semantic allocation is performed independently within each patch, meaning that the transport plan is constrained solely by the voxel representations in that region. Consequently, if a patch does not contain high-level semantic information (such as the early visual cortex), the OT solution naturally assigns nearly zero mass to high-level semantic components (see Fig. A9 for examples). To further test effects of fMRI voxel split on the reconstruction performance, we conduct an ablation study on the number of fMRI patches. Note that for these alternative divisions, we first concatenated the voxels from the early visual, midventral, ventral, midlateral, lateral, midparietal, and parietal regions, and then evenly split the concatenated voxels into the specified number of patches. Results (see Tab. A5) show that coarser or finer fMRI splits lead to a drop in performance, which indicates that the fMRI split strategy is crucial for OT-guided alignment.

## A.2 IMAGE RECONSTRUCTION

The proposed Yo'Mind focuses on brain-to-text semantic decoding, and image reconstruction is beyond the scope of this work, as they correspond to different tasks and model designs. Most existing brain-to-image reconstruction methods such as Chen et al. (2024b) adopt a two-stream architecture, where one stream extracts high-level semantic representations and the other preserves low-level perceptual or spatial structures. In other words, if image reconstruction is desired, Yo'Mind can be naturally integrated with an additional stream dedicated to extracting low-level visual features from the fMRI data.

Here we report several fMRI-to-image decoding instances in Fig. A8. Following Chen et al. (2024b), we use L2-regularized linear regression to predict low-level latent representations of images. The predicted latent representations, together with the decoded image captions, are then fed into Stable Diffusion (version 1.4) (Rombach et al., 2022) for reconstruction. Qualitative results demonstrate that this extension enables the reconstructed images to preserve high fidelity to the original stimuli, both in terms of high-level semantic content and low-level perceptual details.

## A.3 RESULTS ON THE DIR DATASET

To further validate the effectiveness and robustness of Yo'Mind, we perform brain-to-text decoding on the DIR dataset (Shen et al., 2019), which is significantly smaller than NSD (Allen et al., 2022). Note that the image stimuli in DIR originate from ImageNet. Therefore, following Chen et al. (2024b), we leverage SMALLCAP (Ramos et al., 2023) to produce the pseudo labels. Results (see

| Method | Subject | ROIs | Dataset | Text | Language Similarity Metrics ↑ | | | |
|---|---|---|---|---|---|---|---|---|
| | | | | | B@4 | METEOR | ROUGE | SPICE |
| MindGPT (Chen et al., 2025a) | 3 | V1-V4, FFA, PPA, LOC | DIR | SMALLCAP | 15.7 | 12.8 | 35.9 | 10.3 |
| MindArt (Chen et al., 2024b) | 3 | V1-V4, FFA, PPA, LOC | DIR | SMALLCAP | 16.0 | 13.1 | 37.2 | 11.3 |
| Yo'Mind (Ours) | 3 | V1-V4, FFA, PPA, LOC | DIR | SMALLCAP | 18.3 | 15.3 | 39.1 | 12.0 |

Table A6: Quantitative results of brain-to-text reconstruction on the DIR dataset.

| Method | Subject | ROIs | Dataset | Text | Language Similarity Metrics ↑ | | | |
|---|---|---|---|---|---|---|---|---|
| | | | | | SSIM | FID↓ | $CLIP_{score}$ | $CLIP_T$@10 |
| SDRecon (Takagi & Nishimoto, 2023a) | 3 | V1-V4, FFA, PPA, LOC | DIR | SMALLCAP | 0.152 | 17.4 | 0.572 | 24.6% |
| MindGPT (Chen et al., 2025a) | 3 | V1-V4, FFA, PPA, LOC | DIR | SMALLCAP | 0.177 | 2.70 | 0.575 | 31.6% |
| MindArt (Chen et al., 2024b) | 3 | V1-V4, FFA, PPA, LOC | DIR | SMALLCAP | 0.223 | 2.67 | 0.622 | 40.3% |
| Yo'Mind (Ours) | 3 | V1-V4, FFA, PPA, LOC | DIR | SMALLCAP | 0.251 | 2.65 | 0.650 | 42.2% |

Table A7: Quantitative results of brain-to-image reconstruction on the DIR dataset. $CLIP_T$@10 denotes 10-way CLIP image-text scores.

Tab. A6) show that our method remains effective even when trained on small-scale fMRI data. We also report the visual reconstruction results on DIR as a reference (see Tab. A7).

## A.4 COMPUTATIONAL COST AND SCALABILITY

In our framework, we use the Sinkhorn algorithm Cuturi (2013) to solve the assignment problem. The Sinkhorn algorithm relies solely on matrix-vector products (with complexity $O(nm)$), allowing efficient GPU implementation. Moreover, given a fixed number of iterations and fMRI patch embeddings, the computational cost of Sinkhorn grows linearly with the total number of image patch and word embeddings $(N + M)$.

