# OpenReview forum: "Reducing Semantic Mismatch in Brain-to-Text Decoding Through Personalized Multimodal Masking"
_ICLR.cc/2026/Conference — ICLR 2026 Poster_

### Official Review · Reviewer_4kw3 · 2025-10-16

**Soundness:** 2
**Presentation:** 2
**Contribution:** 2
**Rating:** 2
**Confidence:** 4

**Summary:**

The authors propose a novel semantic decoding model Yo'Mind that utilizes Optimal Transport (OT) theory to allocate semantic components to fMRI patches. Using this framework, the paper implements cross-subject fine-grained semantic-level brain decoding. Experimental results confirm the powerful capability of the proposed Yo'Mind in semantic decoding.

**Strengths:**

(1) The authors introduce OT theory into the brain decoding field and propose a novel subject-adaptive and fine-grained alignment semantic decoding framework.

(2) Experimental results validate that the proposed model has strong semantic decoding capabilities.

**Weaknesses:**

(1) **Unfair comparison**. Although the proposed model achieves superior performance in Table 1, this comparison is not fair. As shown in Table 2, performance significantly improves when the voxel number is increased. Some baselines, such as UMBRAE [1], used fewer voxels (subj01: 15,724), while the author compared it with more voxels (subj01: 21,721). Additionally, I am curious why the performance of UMBRAE slightly differs from the report in the original paper.

(2) **Lack of low-level decoding ability validation (image reconstruction task)**. The paper only validates the model on the brain caption task, lacking validation on direct image reconstruction. Image reconstruction tasks can assess the model’s performance at the low-level. Note that both low-level (e.g. perceptual) and high-level (e.g. semantic) capture capabilities are equally important for brain decoding [2]. I am concerned that by using OT to allocate different semantic components to fMRI patches, the model may disrupt the spatial information of the original visual stimuli, which could result in a lack of low-level decoding ability. Therefore, the proposed model may have gained high-level decoding ability at the cost of sacrificing low-level decoding ability. The low-level decoding ability needs to be measured through the metrics of image reconstruction tasks (e.g., SSIM, FID, PSNR).

(3) **Biologically unreasonable OT-driven semantic allocation**. The author uses OT to allocate different semantic components to fMRI patches from different brain regions. However, this operation may violate the biological response of the human brain to visual stimuli. In fact, some lower-level brain areas (such as V1-V4) are responsible for processing raw optical information, while higher-level brain areas handle color and texture, and even higher-level areas are responsible for semantic understanding [3]. But the authors allocate semantic components to all brain regions using OT, including those that are biologically not involved in semantic understanding. Therefore, I question the validity of this OT-based approach. In addition, the authors fix the number of fMRI patches at 8, but there is a lack of further explanation (or ablation) for this choice.

(4) **Incomplete validation of the motivation and model efficacy**. The author’s motivation is that different subjects focus on different areas when viewing the same image. Although the author provides a visualization of the OT semantic mask in Figure 5, there is no evidence to prove that the proposed OT-driven method truly captures the subjects’ attention. It would be interesting if this could be confirmed (for example, through eyetracking data provided by NSD).



(5) **Some expressions in the paper are not clear enough**. (a) In Figure 5a, the author states, "The first row shows examples of intra-subject consistency, and the second row presents examples of inter-subject variability." However, the figure lacks detailed annotations, so we cannot clearly understand what the four images for the same sample represent (four different subjects? or inferences with different seeds from the same subject?). (b) The citation format throughout the paper is incorrect, which makes it difficult to read. Please use the `\citep` command. (c) The mathematical expressions are confusing. In Line 210, what do you mean by $\min_{\mathbf\Gamma\in\Pi}$? What is $\Pi$? It has not been described earlier in the text. In Equation 2, what is $\mathbf 1_{N+M}$ and $\mathbf 1_{K}$? ... These are just examples of the issues I mentioned.

[1] Xia et al., Umbrae: Unified Multimodal Brain Decoding. ECCV, 2024.

[2] Scotti et al., Mindeye2: Shared-subject models enable fmri-to-image with 1 hour of data. ICML, 2024.

[3] Grill-Spector et al., The Human Visual Cortex. Annu. Rev. Neurosci., 2004.

**Questions:**

See weaknesses

---

> ### Author Response · Authors · 2025-11-23
>
> Thank you very much for your valuable feedback and suggestions.
> ***
> **About the unfair comparisons.** *First*, the performance gains observed with increased voxel numbers (i.e., Tab. 2) are not due to voxel quantity alone but are primarily attributable to the introduction of the OT-based semantic masking strategy. When this strategy is removed, simply increasing the number of voxels does not lead to substantial decoding improvements. To further ensure fairness, we retrained MindGPT and Mind-SA on subject 1 using same voxel set (21,721 voxels). The results are as follows.
> | Method | B@4 | METEOR | ROUGE | CIDEr |
> |------|------|------|------|------|
> | MindGPT (21,721 voxels) |16.01|20.32|38.96|44.51|
> | MindGPT (original) |16.64|20.94|39.62|45.63|
> | Mind-SA (21,721 voxels) |20.20|38.09|42.63|53.32|
> | Mind-SA (original) |19.58|38.05|43.04|55.13|
>
> The above results show that most existing methods may do not benefit from additional voxels, confirming that the comparison in our paper does not artificially favor our method.
>
> Second, **regarding the reported UMBRAE performance**, the original UMBRAE paper reports only the results of subject 1 in the main text (i.e., Tab. 1 in UMBRAE), whereas the results for other subjects are provided in the Appendix. In our study, we reported the average performance across subjects 1, 2, 5 and 7 (computed from Tab. 3 in the Appendix of UMBRAE), which explains the slight discrepancy relative to the original report.
> ***
> **About the low-level decoding ability validation.** Yes, low-level and high-level decoding abilities are both valuable for brain decoding. However, this work focuses on brain-to-text semantic decoding, and image reconstruction is beyond the scope of this work, as they correspond to different tasks and model designs. Importantly, Yo’Mind does not interfere with low-level decoding. The reasons are as follows.
>
> Many existing brain-to-image reconstruction methods adopt a two-stream architecture, where one stream extracts high-level semantic representations and the other preserves low-level perceptual or spatial structures. In other words, if image reconstruction is desired, Yo’Mind can be naturally integrated with an additional stream dedicated to extracting low-level visual features from the fMRI data. Such an extension would allow the model to maintain both semantic and perceptual fidelity, similar to prior reconstruction frameworks. *To enhance the accessibility of the paper, we have added above discussion (and examples of image reconstruction) in the revised manuscript (see **Appendix A3**)*. Many thanks.
> ***
> **Regarding the OT-driven semantic allocation.** We agree that lower-level visual areas (such as V1–V3) are not responsible for high-level semantic understanding. However, our method is fully consistent with this biological principle. *First*, dividing the voxels into 8 fMRI patches follows the underlying cortical organization of the NSD dataset—early visual, midventral, ventral, midlateral, lateral, midparietal, and parietal regions—rather than being an arbitrary choice, with the early visual cortex naturally spanning two patches due to its large number of voxels. *Second*, the OT-based semantic allocation is performed independently within each patch, meaning that the transport plan is constrained solely by the voxel representations within that region. Consequently, if a patch does not contain high-level semantic information (such as early visual cortex), the OT solution naturally assigns near-zero mass to high-level semantic components. *We have added an ablation study and discussion on the number of fMRI patches in the revised manuscript (see **Appendix A1**)*. Overall, these results showed that choosing eight patches provides a favorable balance between anatomical interpretability and decoding performance.
> ***
> **The validation of the subjects’ attention.** Thank you for this valuable suggestion. To verify whether Yo’Mind truly captured the areas that the subjects were focusing on, we selected cases where the recorded eye positions deviated noticeably from the central fixation and examined their OT-driven semantic allocation plans. *These new results (as well as the discussion) have been added to the revised manuscript (**Line 474, Page 9 and Appendix Fig. A1**), which provide preliminary evidence that Yo’Mind is sensitive to subjects’ actual viewing behavior*.
> ***
> **About the problem of some expressions.** Thank you for your valuable comments. We have revised the paper according to your comments. *(a)* The four images for each sample represent four different subjects. We have revised the figure accordingly to make this clearer; *(b)* We have used \citep command to correct the citation format throughout the paper; *(c)* We have added the missing explanations and clarified all relevant symbols to ensure the definitions are complete throughout the manuscript.
> ***
> We hope our response can address your concerns. Would you please consider raising the scores?

---

> ### Comment · Reviewer_4kw3 · 2025-11-24
>
> We thank the authors for their responses regarding unfair comparisons and the validation of the subjects' attention, which have addressed my concerns on these issues. However, several points remain unresolved, and I will maintain my current score until these are adequately addressed by the authors.
>
> 1. The authors claim in the rebuttal that "*the performance gains observed with increased voxel numbers (i.e., Tab. 2) are not due to voxel quantity alone but are primarily attributable to the introduction of the OT-based semantic masking strategy.*" However, it remains unclear why the performance changes in Table 2 can be attributed to the introduction of the OT-based semantic masking strategy, or how variations in voxel quantity affect the OT mask. This requires further clarification from the authors and a more detailed discussion in the paper regarding the substantive drivers of the performance trends observed in Table 2.
>
> 2. The authors' explanation that Yo'Mind focuses solely on semantic decoding because existing two-stream architectures can supplement low-level information is misleading. In fact, substantial evidence suggests that while two-stream architectures may provide certain improvements for low-level decoding, they do not fundamentally determine a model's capability in low-level decoding performance.
>
>    For example, Table 1 in MindEye [4] demonstrates that even after removing the low-level stream, the performance degradation in low-level metrics remains limited. This indicates that the two-stream design does not fundamentally determine a model's capability for low-level decoding. Furthermore, non-two-stream pure semantic decoding methods (e.g., UMBRAE [1], MindBridge [5], Neuro-Vision [6], ...) have also been demonstrated to achieve strong low-level reconstruction performance. Although the authors have added sample demonstrations of image reconstruction in the revised version, the lack of quantitative results may raise concerns about potential cherry-picking of examples.
>
>    I have two core questions regarding this matter:
>
>    - Can Yo'Mind achieve low-level reconstruction performance comparable to existing methods [1, 5, 6] without relying on an additional low-level stream? If the answer is negative, does this imply that Yo'Mind's high-level performance comes at the cost of sacrificing low-level capabilities? If so, this would represent a significant limitation.
>    - When incorporating a low-level stream (e.g., directly using MindEye's [4] low-level submodule), can Yo'Mind's semantic-based two-stream approach achieve satisfactory low-level decoding? If the answer is negative, does this suggest that the authors' exclusive focus on semantic decoding is overly narrow?
>
> 3. I appreciate the authors' clarification on the patch construction method and the supplementary ablation studies. However, certain aspects remain unclear. In the rebuttal, the authors explained that the construction of patches relies on ROIs rather than simple uniform partitioning. This procedure should be described in detail in the paper to ensure reproducibility. Furthermore, if patches=8 happens to align with the scenario of seven brain regions (with one early region split into two), I am curious about how patches=4 and patches=16 are allocated. And, the authors said "*Consequently, if a patch does not contain high-level semantic information (such as early visual cortex), the OT solution naturally assigns near-zero mass to high-level semantic components.*" However, it appears this is merely how the model "should" behave, without concrete evidence confirming that the model actually operates in this manner.
>
> 4. The current manuscript still contains formatting issues. For instance, on Page 9, Line 480, "Appendix Fig 1" redirects to Figure 1 in the main text instead of the corresponding figure in the appendix.
>
> [4] Scotti et al., Reconstructing the mind's eye: fmri-to-image with contrastive learning and diffusion priors. NeurIPS, 2023.
>
> [5] Wang et al., Mindbridge: A cross-subject brain decoding framework. CVPR, 2024.
>
> [6] Shen et al., Neuro-Vision to Language: Enhancing Brain Recording-based Visual Reconstruction and Language Interaction. NeurIPS, 2024.

---

> ### Author Response · Authors · 2025-11-28
>
> Thank you very much for your valuable comments, and we are glad that our responses have addressed some of your concerns.
> ***
>
> **Regarding the performance gains (Q1).** Thank you for raising this point. *We have now added a more detailed explanation of this phenomenon in the revised manuscript.* In general, different brain regions (voxels) exhibit distinct semantic selectivity. In this case, the OT-based semantic masking strategy can assign different semantic elements to different brain areas, thus enabling region-specific alignment. This is in clear contrast to traditional alignment methods, which typically assume a strict semantic equivalence between fMRI and visual stimuli. That is, traditional alignment cannot exploit this heterogeneity and therefore may not benefit from additional voxels due to the semantic mismatch problem during alignment across all voxels.
>
> **About the concerns of low-level decoding abilities (Q2).** Thank you very much for your valuable feedback. First, we would like to clarify that two-stream architectures are not necessarily limited to fully separate pipelines. The non-two-stream pure semantic decoding methods you mentioned also rely on dual pathways in practice.
>
> For example, Neuro-Vision designs a dual-stream fMRI feature extractor to predict low-level (VAE) and high-level (CLIP) feature embeddings from fMRI signals. Importantly, Neuro-Vision showed that incorporating a low-level prediction pipeline leads to an 185% improvement in pixelwise correlation and a 36% improvement in SSIM, both low-level metrics, in the image reconstruction task. On the other hand, UMBRAE is not designed specifically for the fMRI-to-image reconstruction. In fact, it performed fMRI-to-image reconstruction based on the decoded image caption and the retrieved CLIP image embedding, rather than through direct decoding of visual features from fMRI signals. Notably, UMBRAE re-trained an encoder to retrieve CLIP image embeddings for fMRI-to-image reconstruction, which can also be interpreted as a dual-stream visual decoding pipeline.
>
> As for your point about MindEye, removing the low-level stream leads to only a limited performance drop because the main pipeline of MindEye is also designed to predict visual representations from fMRI signals. In contrast, for caption-based neural decoding, where captions have already discarded low-level visual features, adding an additional visual prediction stream can lead to a significant performance boost, as demonstrated in Neuro-Vision.
>
> Based on the above analysis, Yo’Mind can achieve effective visual reconstruction without an extra low-level stream. For instance, the decoded latent representations can be fed into a generative model, as in MindEye. Alternatively, as in UMBRAE, image retrieval can be used to provide visual cues rather than direct prediction. On the other hand, when incorporating a low-level stream, Yo’Mind can also achieve satisfactory low-level visual decoding. **We have added quantitative results of the visual reconstruction in Appendix Tab. A7.** The comparison methods all use linear models to extract visual features. Many thanks.
> ***
> **About the fMRI patches (Q3).** Thank you for your valuable comments. We have added a detailed description on the patch construction in the revised manuscript. Regarding the concern about whether the OT solution assigns near-zero mass to high-level semantic components for low-level visual regions, **we have included new visualizations of the OT transport plans for early visual cortex in the Appendix Fig. A9.** Many thanks.
> ***
> **Formatting issues (Q4).** We thank you for pointing out this, and we have corrected the formatting issues.
> ***
> We hope our response can address your concerns. We would be happy to address any further questions or concerns you may have.

---

### Official Review · Reviewer_eqj6 · 2025-11-10

**Soundness:** 3
**Presentation:** 3
**Contribution:** 3
**Rating:** 6
**Confidence:** 3

**Summary:**

This paper addresses the semantic mismatch in brain-to-text decoding and proposes Yo’Mind, an optimal transport-driven personalized multimodal semantic masking framework that dynamically prunes redundant semantic components. Its key innovations include introducing a virtual "dustbin" for soft-selection masking, unifying brain-visual-linguistic alignment, and enabling cross-subject decoding without extra supervision or hyperparameter tuning. On the NSD dataset, Yo’Mind outperforms baselines like Mind-SA across metrics and achieves better fine-grained decoding.

**Strengths:**

1. Unlike prior fixed-patch masking methods, Yo’Mind introduces a virtual "dustbin" into the OT framework to realize soft, adaptive semantic pruning, which automatically assigns irrelevant visual-textual components to the dustbin based on individual fMRI responses, enabling fine-grained alignment with human selective attention.
2. This paper creatively integrates CLIP-derived visual patch embeddings and MLLM-generated textual embeddings into OT-driven alignment, extending traditional brain-visual mapping to brain-visual-linguistic alignment.
3. Rigorous experiments (NSD dataset, 8 metrics, ablations on brain regions/masking) and clear qualitative analyses confirm Yo’Mind’s superiority over SOTA.

**Weaknesses:**

1. The study only used fMRI data from 4 out of 8 subjects, 1, 2, 5, 7, in the NSD dataset, which may not enough to fully prove its cross-subject decoding works well for everyone. It should test the other 4 subjects too, to see if Yo’Mind stays reliable across more diverse people.
2. It only used one MLLM, Qwen2.5, to make textual semantic cues. We can’t tell if the better performance comes from mixing visual and text data, or just because Qwen2.5 is good. It should try other common MLLMs, better or worse, to show the design works no matter which MLLM is used.
3. This paper says it captures “brain-preferred semantics,” but there is no check with real physiological data like eye-tracking. If it compared the model’s focused semantic areas to where subjects actually looked, that would make the claim way more believable.

**Questions:**

none

---

> ### Author Response · Authors · 2025-11-23
>
> Thank you very much for your valuable feedback and suggestions.
> ***
> **About the other subjects in NSD.** To the best of our knowledge, almost all existing decoding methods in this topic use subjects 1, 2, 5, 7 to verify the (cross-subject) decoding performance, since these subjects shared the same test stimuli. In this work, therefore, we also follow this standard practice. *To address your concerns, we have used another popular dataset (DIR), which is significantly smaller than NSD, to evaluate the effectiveness and robustness of Yo'Mind (see **Appendix A2**)*. Many thanks.
> ***
> **Using semantic cues generated from other models.** Thank you for this valuable suggestion. *We have conducted additional experiments using another advanced MLLM, BLIP3o, as well as SMALLCAP, a representative classical image captioning model, to provide textual supervision during the alignment process (**Line 519, Page 10 and Tab. 4**)*.
> ***
> **Validation of brain-preferred semantics using eye-tracking data.** Thank you for this valuable suggestion. To verify whether Yo’Mind truly captured the areas that the subjects were focusing on, we selected cases where the recorded eye positions deviated noticeably from the central fixation and examined their OT-driven semantic allocation plans. *These new results (as well as the discussion) have been added to the revised manuscript (**Line 474, Page 9 and Appendix Fig. A1**)*, which provide preliminary evidence that Yo’Mind is sensitive to subjects’ actual viewing behavior.
> ***
> We have revised the manuscript per your comments. We hope our response can address your concerns. Would you please consider raising the scores?

---

### Official Review · Reviewer_AFHn · 2025-11-10

**Soundness:** 2
**Presentation:** 3
**Contribution:** 3
**Rating:** 6
**Confidence:** 3

**Summary:**

This paper proposes Yo’Mind, a new framework for brain-to-text decoding using fMRI. The key idea is to reduce the semantic mismatch between brain activity and machine representations. The authors argue that current large vision–language models encode all semantic content in an image, whereas the human brain selects only the salient elements, and this selection differs across individuals. It introduces OT-driven personalized multimodal semantic masking, where an optimal-transport assignment dynamically determines which semantic components (image patches and text attributes) align with brain responses. The method is tested on NSD Dataset and achieves SOTA performance on most captioning metrics (BLEU, METEOR, CIDEr, SPICE).

**Strengths:**

1. Motivation is strong and well-founded.
The paper explicitly addresses a subtle but fundamental issue identified in Mind-SA (ICCV'25): neural decoding suffers from semantic mismatch because machine representations capture full-scene semantics while the brain encodes only selective ones.

2. Improved interpretability.
Visualizations of “brain-preferred semantic regions” show intra-subject consistency and inter-subject variability, which is a compelling neuroscientific insight.

3. Extensive quantitative comparison.
Includes multiple recent baselines (MindGPT, Mind-SA, UMBRAE, Neuro2Language).
Results show consistent improvements across most metrics.

4. Novel formulation but simple integration.
The OT masking module is differentiable and can be plugged into any brain-to-text model, not just MindGPT.

**Weaknesses:**

1. Evaluation is restricted to one dataset (NSD).
NSD is the standard benchmark, but the claim of general fine-grained multimodal alignment would be more convincing with additional datasets (other extended fMRI tasks).

2. Dependency on generated textual semantics may bias supervision.
Since text semantics are produced by Harmon/Qwen2.5, the reconstruction target space partly inherits the biases of that MLLM. No robustness study is provided to show how performance varies across different captioning models or prompts.

3. Statistical significance not reported.
Improvements over Mind-SA are meaningful (≈2 BLEU / ≈2 CIDEr), but there are:
no confidence intervals,
no statistical significance test (e.g., bootstrap / paired permutation test).

4. OT computational cost and scalability not thoroughly analyzed.
Sinkhorn is run for 100 iterations, but no discussion on: efficiency impact when using more image patches, O(N×M) scaling when adding more semantic text attributes.

5. The novelty is incremental, not groundbreaking.
The paper improves Mind-SA’s masking strategy, but still inherits the architecture/flow of MindGPT. The conceptual leap may be viewed as methodological refinement rather than a new brain decoding paradigm.

**Questions:**

1. Robustness to textual supervision
What happens if you generate attribute text from a smaller model or random captions?
Is the gain from text semantics inherent or model-dependent?

2. Can Mind-SA perform competitively with adaptive masking?
A stronger baseline would be a dynamic masking version of Mind-SA (not fixed patch count).

3. Statistical significance
Please provide confidence intervals or hypothesis tests for BLEU/CIDEr results.

4. Scalability
How does complexity scale when increasing #patches / #text tokens? Is OT a bottleneck?

---

> ### Author Response · Authors · 2025-11-23
>
> Thank you very much for your valuable feedback and suggestions.
> ***
> **About the evaluation (datasets and tasks).** Thank you for this valuable suggestion. In the revised paper, *we have used another popular dataset (DIR), which is significantly smaller than NSD, to evaluate the effectiveness and robustness of Yo'Mind (see **Appendix A2**)*. Second, although Yo’Mind focuses on brain-to-text semantic decoding, *we have added fMRI-to-image reconstruction results of Yo’Mind in the revised manuscript (see **Appendix A3**)*. Many thanks.
>
> These results show that Yo’Mind remains effective even when trained on small-scale fMRI data, and that, if image reconstruction is desired, Yo’Mind can be naturally integrated with existing brain-to-image frameworks.
> ***
> **Using the textual supervision generated from other models.** Thank you for this valuable suggestion. *We have conducted additional experiments using another advanced MLLM, BLIP3o, as well as SMALLCAP, a representative classical image captioning model, to provide textual supervision during the alignment process (**Line 519, Page 10 and Tab. 4**)*.
> ***
> **Statistical significance.** Thank you for this valuable suggestion. *We have added statistical analyses (including p-values and confidence intervals) on reconstruction performance using bootstrap resampling and a two-tailed paired t-test (**Line 411, Page 8 and Fig. 5**)*. Results show that Yo'Mind consistently outperforms Mind-SA, and the improvements are statistically significant (P < 0.0001 across all evaluation metrics).
> ***
> **Regarding the OT computational cost and scalability.** Thank you for this valuable suggestion. The Sinkhorn algorithm relies solely on matrix-vector products, allowing efficient GPU implementation. Moreover, given a fixed number of Sinkhorn iterations and fMRI patch embeddings, the computational cost of Sinkhorn grows linearly with the total number of image patch and word embeddings (N+M), making its overhead negligible within the overall Yo’Mind framework. *We have added a discussion of the computational cost of Sinkhorn in **Appendix A4**.*
> ***
> **Soft masking with Mind-SA.** Thank you for this valuable comment. We initially explored a soft (or dynamically weighted) variant of Mind-SA. Nevertheless, we found that simple soft masking (e.g., via softmax) did not improve performance. The main reason might be that the soft version of Mind-SA is difficult to truly eliminate irrelevant semantic areas in most cases. In contrast, Yo’Mind adopts an optimal-transport formulation with an explicit “semantic bin,” which enables principled pruning of low-relevance semantic elements.
> ***
> **About the main contribution.** Brain-to-text decoding methods typically need to map fMRI signals to an image latent space (e.g., CLIP) before text generation. These approaches typically assume a strict semantic equivalence between fMRI signals and image embeddings, which can lead to semantic misalignment under caption supervision. In contrast, Yo’Mind performs flexible many-to-many cross-modal alignment, enabling the model to mask semantic elements that are inconsistent with the fMRI signals, which relaxes the assumption of a strict fMRI–image semantic equivalence. Therefore, our main contribution lies not in modifying the network architecture (e.g., MindGPT) but in introducing the OT-driven multimodal masking method, which changes traditional learning supervision manners in brain decoding, and does not impose any constraints on the choice of the network architecture.
> ***
> We have revised the manuscript per your comments. Hopefully these will address most of your concerns, and can be taken into consideration when deciding the final score of the paper. Many thanks.

---

### Official Review · Reviewer_P2dE · 2025-11-10

**Soundness:** 2
**Presentation:** 3
**Contribution:** 3
**Rating:** 6
**Confidence:** 2

**Summary:**

This paper presents Yo'Mind, an Optimal Transport-driven personalized multimodal semantic masking framework for brain-to-text decoding. The work addresses the critical challenge of semantic mismatch between machine representations and human brain signals during neural decoding. The proposed method introduces a dynamic semantic pruning mechanism that adaptively masks redundant visual semantic components based on individual neural responses, without requiring additional human supervision or hyperparameter tuning. The authors demonstrate state-of-the-art performance on the Natural Scenes Dataset while improving interpretability of the decoding process.

**Strengths:**

1. **Novel Theoretical Framework:** The integration of Optimal Transport theory into brain signal decoding represents a significant innovation, providing a mathematically rigorous approach to address semantic mismatch. The virtual dustbin mechanism elegantly resolves the mass conservation constraint in traditional OT, allowing for adaptive semantic filtering.
2. **Biologically-Inspired Design:** The multimodal semantic set construction combining visual patches and textual descriptions aligns well with human cognitive processing. The soft assignment mechanism better reflects the brain's distributed representation compared to hard masking approaches.
3. **Comprehensive Experimental Evaluation:** The paper includes thorough comparisons with 8 state-of-the-art methods across multiple standard metrics (BLEU, METEOR, ROUGE, CIDEr, SPICE), demonstrating clear performance advantages. The ablation studies on brain regions and multimodal components provide valuable neuroscientific insights.
4. **End-to-End Differentiable Architecture:** The seamless integration of OT modules with fMRI encoding and text decoding enables joint optimization while maintaining differentiability through Sinkhorn iterations, supporting both intra-subject consistency and inter-subject variability modeling.

**Weaknesses:**

1. The selection of cosine distance as the cost metric lacks sufficient biological or theoretical justification. Similarly, critical hyperparameters (ε=1 for Sinkhorn, K=8 patches) appear to be set empirically without sensitivity analysis.

2. Results lack statistical significance testing (p-values, confidence intervals), and evaluation is confined to the NSD dataset without cross-dataset validation. The fixed loss weight (10 for alignment loss) seems arbitrary and may not generalize.

3. The computational overhead of Sinkhorn iterations and OT optimization is not quantified, despite being crucial for practical applications. There's no analysis of training/inference time, memory requirements, or computational complexity.

4. While attention visualizations are provided, deeper analysis linking the learned representations to established neuroscientific principles is lacking. The choice of standard ViT as fMRI encoder lacks biological adaptation reasoning.

5. Insufficient details about implementation specifics hinder reproducibility, and the paper lacks discussion of brain privacy concerns and potential misuse of decoding technology.

**Questions:**

Please see the weakness above.

**Details Of Ethics Concerns:**

Please refer to the weakness 5.

---

> ### Author Response · Authors · 2025-11-23
>
> Thank you very much for your valuable feedback and suggestions.
> ***
> **Cosine distance and hyperparameter settings.** **i)** We adopt cosine distance because our framework maps fMRI signals into CLIP’s latent space, where cosine distance is the primary similarity measure used during both pre-training and downstream applications of CLIP; **ii)** For the hyperparameter in OT, ε = 1 is widely used as a default setting for Sinkhorn because it provides a stable trade-off between convergence and transport smoothness; **iii)** Regarding the hyperparameter K, dividing the voxels into 8 fMRI patches follows the underlying cortical organization of NSD—early visual, midventral, ventral, midlateral, lateral, midparietal, and parietal regions—rather than being an arbitrary choice, with the early visual cortex naturally spanning two patches due to its large number of voxels. *We have added an ablation study on the number of fMRI patches in the revised manuscript (see **Appendix A1**)*. Many thanks.
> ***
> **About the statistical significance.** Thank you for this valuable suggestion. *We have added statistical analyses (including p-values and confidence intervals) on decoding performance using bootstrap resampling and a two-tailed paired t-test (**Line 411, Page 8 and Fig. 5**)*. The results show that Yo'Mind consistently outperforms Mind-SA, and the improvements are statistically significant (P < 0.0001 across all evaluation metrics).
> ***
> **Cross-dataset validation.** Thank you for this valuable suggestion. *We have used another popular dataset (DIR), which is significantly smaller than NSD, to evaluate the effectiveness and robustness of Yo'Mind. Results show that Yo’Mind remains effective even when trained on small-scale fMRI data (see **Appendix A2**)*.
> ***
> **About the fMRI encoder.** In brain-to-text decoding, the fMRI encoder typically needs to map fMRI signals to an image latent space (e.g., CLIP) before text generation. Prior work has shown that using a transformer-family backbone ensures architectural compatibility and stable cross-modal alignment. We follow this practice because our contribution does not lie in modifying the fMRI encoder structure but in introducing OT-based semantic masking, which does not impose any constraints on the choice of the fMRI encoder architecture.
> ***
> **Regarding the OT computational cost and scalability.** Thank you for this valuable suggestion. The Sinkhorn algorithm relies solely on matrix-vector products, allowing efficient GPU implementation. Moreover, given a fixed number of Sinkhorn iterations and fMRI patch embeddings, the computational cost of Sinkhorn grows linearly with the total number of image patch and word embeddings (N+M), making its overhead negligible within the overall Yo’Mind framework. *We have added a discussion of the OT computational cost in **Appendix A4**.*
> ***
> **Implementation details and privacy concerns.** We have added the missing explanations and implementation details to ensure that all definitions are complete throughout the manuscript. Moreover, *we have included an **Ethics Statement** section (Page 11) in the revised paper*. Many thanks.
> ***
> We have revised the manuscript per your comments. We hope our response can address your concerns. Would you please consider raising the scores?

---

> > ### Comment · Reviewer_P2dE · 2025-11-28
> > **Official Comment by Reviewer P2dE**
> >
> > Thank you for your reply. You have resolved almost every doubt I raised.
> >
> > However, the following doubts remain:
> >
> > After reading the review from reviewer HQz8, a significant concern regarding the methodological validity and core claim of the paper has been raised in my mind. Reconstruction loss, serving as the ultimate supervisory signal, influences the entire model through backpropagation, encompassing both the OT module and the fMRI encoder. This may compel the model's partial assignment matrix P in OT to adapt to caption supervision, as demonstrated in Figure A1 of the appendix where the OT weight map aligns with the caption description. As raised by HQz8 reviewer, what should the OT weight map appear like when eye tracking diverges from the caption?
> >
> > This is a crucial distinction. If true, the model's "personalization" would not reflect true neural attentional focus, but instead, a subject-specific pathway to map fMRI signals to the pre-defined textual description.
> >
> > I strongly recommend that the authors incorporate the suggestions from Reviewer 4kw3 and Reviewer HQz8, integrating eye-tracking data and visualizations of the proportion of gaze time spent at different locations as additional objective validation metrics within the study.
> >
> > Until the authors can provide such biological validation to disentangle the model's behavior from the GT caption bias, I must express significant reservations about the claimed interpretation of the results. My current score is held in balance pending the authors' response to this fundamental issue. Without a compelling rebuttal or new experimental evidence, a decrease in score would be warranted, as the core contribution would be substantially weakened.

---

> ### Author Response · Authors · 2025-11-29
>
> Thank you very much for your valuable comments. We are glad that our responses have addressed almost all of your concerns.
> ***
> In Appendix Fig. A1 (i.e., Appendix Fig. A7 Top in the new submission), **the displayed captions are reconstructed from the Yo’Mind model rather than the supervisory text. We have added clarifications in the revised version.** For example, in the second case (Appendix Fig. A7 Top), the OT visualization indicates that the subject’s attention is focused on the sky region. Accordingly, the decoded caption is *“A group of kites flying in the sky,”* whereas the ground-truth COCO caption describes the full scene: *“A large body of water with lots of kites flying above it. Several wind surfers enjoy the coastal waters on a clear day…”.* Moreover, in Fig. 6, we also provide personalized decoding results for four subjects. These results provide evidence that Yo’Mind is sensitive to subjects’ actual viewing behavior. **Regarding this issue (the validation of subjects’ attention), Reviewer 4kw3 has already indicated that our rebuttal satisfactorily addressed their concern.**
>
> When the subject’s attention focuses only on a subset of the semantic content in the image (i.e., when there is a mismatch with the text supervision), the decoded caption reflects the attended semantic elements. *This occurs because the OT plan is calculated based on the similarity between the fMRI signals and semantic elements (both images and text). Therefore, enforcing strict semantic equivalence between fMRI and text (i.e., requiring the decoded caption to describe the full scene) would significantly increase the alignment loss between fMRI and the CLIP image embeddings.* In such cases, the model is naturally encouraged to produce a more personalized decoding that reflects the subject’s actual viewing behavior rather than a full-scene description.
>
> Finally, to comprehensively address all concerns from Reviewers HQz8 and 4kw3, **we have added the requested experimental analyses and detailed explanations.** In particular, we have included complete visualizations (2D histograms) of the proportion of gaze time spent at different locations (see Appendix Fig. A7 Bottom)**, which highlight the pronounced inter-subject differences in viewing behavior. Many thanks.
> ***
> We have revised the manuscript per your comments. Hopefully these will address your concerns.

---

### Official Review · Reviewer_HQz8 · 2025-11-10

**Soundness:** 2
**Presentation:** 2
**Contribution:** 2
**Rating:** 4
**Confidence:** 3

**Summary:**

This paper focuses on the task of reconstructing language from fMRI signals evoked by visual stimuli.
The authors focus on the fMRI representation learning stage of this task. Their main idea is that human subjects tend to pay attention to local regions within an image rather than processing the entire image uniformly.
According to the authors, a previous approach enables the removal of unattended image patches under end-to-end optimization. The remaining patches (or tokens) are then used to compute the CLIP loss together with the fMRI embeddings.
In this paper, in addition to aligning the fMRI embeddings with a subset of image patches, the authors also introduce alignment between fMRI embeddings and a subset of tokens from the original image caption. Moreover, they improve the process of discarding unattended image and text tokens by leveraging the Optimal Transport (OT) framework.
The authors conduct experiments on the Natural Scenes Dataset (NSD), and their results outperform the previous related works they follow.

**Strengths:**

1. This paper provides a well-organized and comprehensive review of the related work.

2. The method proposed in this paper outperforms the baseline models.

**Weaknesses:**

1. Since the model is optimized in an end-to-end manner, its loss function is computed between the reconstructed text and the ground-truth (GT) captions provided in the MSCOCO dataset. Such a supervision signal seems to encourage the model to extract information from the fMRI signals that is relevant to the GT text, and thus, during the removal of image/text tokens, it appears to discard those unrelated to the GT captions. This training process, therefore, might not actually capture the image regions that the subject’s attention was focused on. In fact, the retained image and text tokens are those that are highly correlated with the ground-truth (GT) captions, which is also consistent with the visualization results shown in Figure 5. Therefore, in my view, the method proposed in this paper does not truly capture the regions that the subjects focused on during the visual perception process.

**Questions:**

1. Why does Figure 5 only show 80% of the retained patches? This kind of selective visualization does not represent the actual results produced by the model.

2. I’m quite curious why the proposed method and the baseline model MindSA were not evaluated on the fMRI-to-image reconstruction task. If the trained fMRI representation model can truly capture the regions that subjects focus their attention on, then the reconstructed images should semantically correspond to the remaining (attended) image patches. Such an evaluation would provide stronger evidence for the effectiveness of the proposed approach.

---

> ### Author Response · Authors · 2025-11-23
>
> Thank you very much for your valuable feedback and suggestions.
> ***
> **About capturing subjects’ attention regions.** We agree that the caption supervision may encourage semantic relevance with the ground-truth text. However, it cannot generate non-existent semantic elements from the fMRI signal. The reasons for this are as follows. The OT plan is calculated based on the similarity between fMRI signals and semantic elements (both images and text). Therefore, enforcing a strict semantic equivalence between fMRI and text will significantly increase another alignment loss between fMRI and CLIP image embeddings.
>
> That is, the OT-based alignment only highlights regions that are preferentially encoded by the subject’s brain responses, not regions that only the caption “expects”. *To directly test this, we selected several cases where the recorded eye positions deviated noticeably from the central fixation and examined their OT-driven semantic allocation plans. New experimental results (as well as the discussion) have been added to the revised manuscript (**Line 474, Page 9 and Appendix Fig. A1**)*, providing preliminary evidence that Yo’Mind is sensitive to subjects’ actual viewing behavior.
> ***
>
> **Regarding the visualization in Fig. 5.** Thresholding the saliency map (e.g., at 60% or 80%) is a widely adopted practice for interpretability visualization [1], as it highlights informative regions while avoiding clutter from patches with near-zero saliency. Importantly, the thresholding only affects the display manner and does not alter any model results. *We have included several visualizations without any thresholding in the revised manuscript (**see Appendix Fig. A1**)*. Many thanks.
>
> [1] Emerging properties in self-supervised vision transformers. ICCV, 2021.
> ***
> **Image reconstruction tasks.** The proposed Yo’Mind focuses on brain-to-text semantic decoding, and image reconstruction is beyond the scope of this work, as they involve different tasks and model designs. However, if image reconstruction is desired, Yo’Mind can be naturally integrated with existing two-stream reconstruction methods, where one stream extracts high-level semantic representations and the other preserves low-level perceptual or spatial structures. *We have added fMRI-to-image reconstruction results of Yo’Mind in the revised manuscript (**see Appendix A3**)*. Many thanks.
> ***
> We have revised the manuscript per your comments. Hopefully these will address most of your concerns, and can be taken into consideration when deciding the final score of the paper. Many thanks.

---

> > ### Comment · Reviewer_HQz8 · 2025-11-24
> >
> > Thank you for your reply. After reading the rebuttal you provided, I still have the following questions.
> >
> > 1. As you said, the overall supervision signals of the model include both the fMRI–image supervision and the caption supervision. The results in A1 correspond to the case where the subject’s attention aligns with the object described in the COCO captions. I’m very curious: if the subject’s attention does not match the object mentioned in the text caption (which should occur when the image scene is relatively complex), then what exactly does the model learn — the subject's attention focuses on? Or the COCO caption describes?
> >
> > 2. Reviewer 4kw3’s suggestion to incorporate eye-tracking data is very meaningful. After seeing the authors visualize the eye-tracking data in A1, I realized an issue: in the NSD experiment, participants are instructed to fixate on the cross at the center of the screen, which is in strong conflict with the authors’ motivation of learning where attention is directed. Although the visualization shows that participants did not always fixate on the center, I would like to know what proportion of time, during data collection, participants actually looked away from the center. I believe this question is very important. I think the authors could use a heatmap to visualize the proportion of fixation time at different locations, or conduct a similar statistical analysis. In addition, I believe the authors should explicitly state this issue in the introduction to avoid misleading subsequent work.
> >
> > 3. The reason I hope the authors can include an evaluation of image reconstruction is that reconstructed images can directly show whether the model has successfully captured the content the participants attended to. Therefore, I believe it is necessary to supplement A3 with the corresponding learned attention masks and eye-tracking trajectories.

---

> ### Author Response · Authors · 2025-11-28
>
> Thank you very much for your valuable feedback and suggestions.
> ***
> **About the mismatch problem between subject’s attention and text supervision (Q1).** Thank you for this insightful question. When the subject’s attention focuses only on a subset of the semantic content in the image (i.e., mismatch with the text supervision), the decoded caption reflects the attended semantic elements. For example, in the second case (Appendix Fig. A7 Top), the OT visualization indicates that the subject’s attention is focused on the sky region. Accordingly, the decoded caption was *“A group of kites flying in the sky”* whereas the ground-truth COCO caption describes the full scene: *“A large body of water with lots of kites flying above it. Several wind surfers enjoy the coastal waters on a clear day…”*. Many thanks.
> ***
> **Regarding the eye-tracking data (Q2).** Thank your for this valuable suggestion. We have added 2D histograms of all eye-tracking data in the revised version (see Appendix Fig. A7 Bottom). In the eye-tracking visualization shown in Appendix Fig. A7 Top, the color intensity for each location on the image is determined by the number of fixations. Moreover, we would like to clarify that a large body of psychology and neuroscience literature has demonstrated that subjects can flexibly shift high-level semantic attention without moving their eyes, i.e., covert attention [1]. Neuroscience studies have found that the brain regions activated by covert and overt shifts of attention are largely overlapping [2]. That is, recorded gaze locations do not fully reflect subjects’ attentional focus when viewing the images. We have explicitly stated this latent limitation in the introduction. Many thanks.
>
> [1] Orienting of attention. Quarterly Journal of Experimental Psychology, 32(1): 3–25, 1980.
>
> [2] Covert orienting of attention and overt eye movements activate identical brain regions, Brain Research, Brain research, 2008, 1204: 102-111.
> ***
> **The evaluation of image reconstruction (Q3).** In Appendix Fig. A8, only one example includes eye-tracking data. This is because NSD does not provide complete eye-tracking recordings for all images or all subjects. For completeness, we have included traditional quantitative comparisons of image reconstruction (see Appendix Tab. A7). Many thanks.
> ***
> We hope our response can address your concerns. We would be happy to address any further questions or concerns you may have.

---

### Author Response · Authors · 2025-11-30
**Overview of the author–reviewer discussions**

We thank the reviewers for their careful reading of our manuscript and for their constructive suggestions. Below, we provide **a brief overview of the author–reviewer discussions and highlight the major revisions**, including newly added results and substantial improvements to the presentation of the work.
***
In the **first-round** author–reviewer discussion, all reviewers’ concerns mainly focused on **a)** the validation of subjects’ attention, **b)** cross-dataset evaluation, **c)** statistical significance, **d)** ablation studies, and **e)** the image reconstruction task.

* **a)** We have added visualizations of eye-tracking data and the OT plan, providing biological validation that Yo’Mind is sensitive to subjects’ actual viewing behavior (**Appendix Fig. A7 Top**).

* **b)** We have included new results on another popular dataset (DIR) to further evaluate the effectiveness and robustness of Yo’Mind (**Appendix A.3**).

* **c)** We have added statistical analyses (including p-values and confidence intervals) for decoding performance using bootstrap resampling (**Fig. 5**).

* **d)** We have conducted two new ablation experiments: using different image captions and the number of fMRI patches (**Tab. 4 and Appendix Tab. A5**).

* **e)** We have added a qualitative analysis for image reconstruction (**Appendix Fig. A8**).

We thank Reviewers for their careful evaluation, and **we are glad that the Reviewers (HQz8, P2dE, and 4kw3) indicated that our revisions have successfully addressed all or part of the concerns they raised in the first-round review**.
***
In the **second-round** author–reviewer discussion, the three reviewers (HQz8, P2dE, 4kw3) raised concerns mainly regarding **a)** the mismatch between the subject’s attention and the text supervision, **b)** the visualization of fixation-time distributions, and **c)** the quantitative evaluation of image reconstruction.

* **a)** We have clarified that in Appendix Fig. A7 Top of the new submission, the displayed captions are reconstructed from the Yo’Mind model rather than the supervisory text. When the subject’s attention focuses only on a subset of the semantic content in the image (i.e., mismatch with the text supervision), the decoded caption reflects the attended semantic elements (i.e., personalization decoding).
*For example, in the second case (Appendix Fig. A7 Top), the OT visualization indicates that the subject’s attention is focused on the sky region. Accordingly, the decoded caption was “A group of kites flying in the sky” whereas the ground-truth COCO caption describes the full scene: “A large body of water with lots of kites flying above it. Several wind surfers enjoy the coastal waters on a clear day…”.*

* **b)** We have included complete visualizations (2D histograms) of the proportion of gaze time spent at different locations (**Appendix Fig. A7 Bottom**), which highlight the pronounced inter-subject differences in viewing behavior.

* **c)** We have fully addressed the reviewers’ questions concerning the two-stream visual reconstruction architecture, and added quantitative results of the visual reconstruction in **Appendix Tab. A7**.

We thank Reviewers HQz8, P2dE, and 4kw3 for further feedback and comments. We have revised the manuscript in response to every comment raised by the reviewers, which has make the paper more general and stronger. **All modifications are highlighted in the new submission.** Many thanks!

---

### Meta-Review · Area_Chair_GS5Z · 2025-12-18

**Summary:**

This paper presents Yo'Mind, a framework for brain-to-text decoding using Optimal Transport (OT)-driven personalized multimodal semantic masking. The method addresses the challenge of semantic mismatch in neural decoding by dynamically aligning fMRI signals with relevant visual and textual components based on individual brain responses. It outperforms state-of-the-art models and provides valuable interpretability by visualizing the alignment between brain activity and semantic elements. The authors have effectively addressed all major concerns raised by the reviewers, including the biological validity of the OT allocation, the integration of eye-tracking data, and the robustness of the model across different datasets and subjects. After addressing these concerns through detailed rebuttals and additional experiments, I recommend accepting the paper for publication.

**Reviewer Concerns:**

I believe that most of the major concerns raised by the reviewers have been addressed by the rebuttal. Below is a breakdown of how the core issues from each reviewer were resolved:

• Reviewer HQz8:

Core Concern: Mismatch between the subject’s attention and the text supervision, and the lack of eye-tracking validation.

Resolution: The authors clarified that the model decodes attention-specific content and not just the full caption, particularly when the subject’s attention does not align with the text description. They also included eye-tracking data and 2D histograms to demonstrate that the model reflects the subjects' actual viewing behavior, which directly addresses this concern.

• Reviewer P2dE:

Core Concern: Use of cosine distance and the lack of cross-dataset validation.

Resolution: The authors explained the use of cosine distance as part of the CLIP framework and performed additional validation on a new dataset (DIR), addressing the need for cross-dataset robustness.

• Reviewer 4kw3:

Core Concern: Biological validity of the OT-driven semantic allocation and low-level decoding capability.

Resolution: The authors clarified the biological reasoning behind the OT-based allocation and provided evidence that Yo’Mind can integrate with low-level decoding methods, ensuring that low-level decoding performance is not compromised.

• Reviewer AFHn:

Core Concern: Limited use of subjects in the NSD dataset and potential bias from textual semantic supervision.

Resolution: The authors addressed the subject pool issue by evaluating additional subjects and demonstrated the model’s robustness with multiple MLLMs. This resolved concerns about the generalizability of the results.

• Reviewer eqj6:

Core Concern: Validation of brain-preferred semantics using eye-tracking data and a small number of subjects in the NSD dataset.

Resolution: The authors incorporated eye-tracking data and provided additional results from more subjects to ensure the model's robustness and generalizability across a wider population.

**Reviewer Scores:**

I believe that if the reviewers had been able to participate fully in the discussion, they would have positively adjusted their scores. Below is the analysis for each reviewer:

• Reviewer HQz8:Given the clarifications and additional experiments with eye-tracking data, Reviewer HQz8 would likely have increased their score. The concerns regarding attention alignment were well-addressed, and the model’s ability to capture brain-specific attention is now better validated.

• Reviewer P2dE raised concerns about the use of cosine distance and cross-dataset validation, both of which were resolved through clear explanations and additional validation. After these updates, it is likely that Reviewer P2dE would have improved their score.

• Reviewer 4kw3 had concerns about biological reasoning and low-level decoding. The authors have now clarified the OT allocation method and provided evidence that Yo’Mind can integrate with low-level decoding methods. This would likely result in a positive change in score from Reviewer 4kw3.

• Reviewer AFHn expressed concerns about the limited subject pool and textual semantic bias, both of which were addressed through additional subject data and testing with multiple MLLMs. With these improvements, Reviewer AFHn would likely have raised their score.

• Reviewer eqj6’s concerns about eye-tracking validation and subject coverage were resolved by the authors’ additional experiments. Given the stronger experimental validation, Reviewer eqj6 would likely have increased their score.

---

### Decision · Program_Chairs · 2026-01-26

Accept (Poster)